# When Semantics Mislead Vision:
# Mitigating Large Multimodal Models Hallucinations in Scene Text Spotting and Understanding

**Yan Shu**[1*]  **Hangui Lin**[3*]  **Yexin Liu**[2*]  **Yan Zhang**[4,5]  **Gangyan Zeng**[6]
**Yan Li**[3†]  **Yu Zhou**[7]  **Ser-Nam Lim**[8]  **Harry Yang**[2]  **Nicu Sebe**[1]

[1] University of Trento  [2] Hong Kong University of Science and Technology
[3] University of International Relations
[4] Institute of Information Engineering, Chinese Academy of Sciences
[5] School of Cyber Security, University of Chinese Academy of Sciences
[6] Nanjing University of Science and Technology
[7] VCIP & TMCC & DISSec, College of Computer Science, Nankai University
[8] University of Central Florida
{yan.shu,niculae.sebe}@unitn.it
https://github.com/shuyansy/MLLM-Semantic-Hallucination

## Abstract

Large Multimodal Models (LMMs) have achieved impressive progress in visual perception and reasoning. However, when confronted with visually ambiguous or non-semantic scene text, they often struggle to accurately spot and understand the content, frequently generating semantically plausible yet visually incorrect answers, which we refer to as semantic hallucination. In this work, we investigate the underlying causes of semantic hallucination and identify a key finding: Transformer layers in LLM with stronger attention focus on scene text regions are less prone to producing semantic hallucinations. Thus, we propose a training-free semantic hallucination mitigation framework comprising two key components: (1) ZoomText, a coarse-to-fine strategy that identifies potential text regions without external detectors; and (2) Grounded Layer Correction, which adaptively leverages the internal representations from layers less prone to hallucination to guide decoding, correcting hallucinated outputs for non-semantic samples while preserving the semantics of meaningful ones. To enable rigorous evaluation, we introduce TextHalu-Bench, a benchmark of 1,740 samples spanning both semantic and non-semantic cases, with manually curated question–answer pairs designed to probe model hallucinations. Extensive experiments demonstrate that our method not only effectively mitigates semantic hallucination but also achieves strong performance on public benchmarks for scene text spotting and understanding.

## 1  Introduction

Scene text, as a self-descriptive visual element, conveys rich semantic information that is crucial for downstream applications such as autonomous driving, product analysis, and assistive technologies. Effectively spotting and understanding scene text [1, 2, 3, 4, 5, 6, 7, 8] thus attracts growing attention from the deep learning community.

---

[*]Equal contribution.
[†]Corresponding author <liyan@uir.edu.cn>.

39th Conference on Neural Information Processing Systems (NeurIPS 2025).

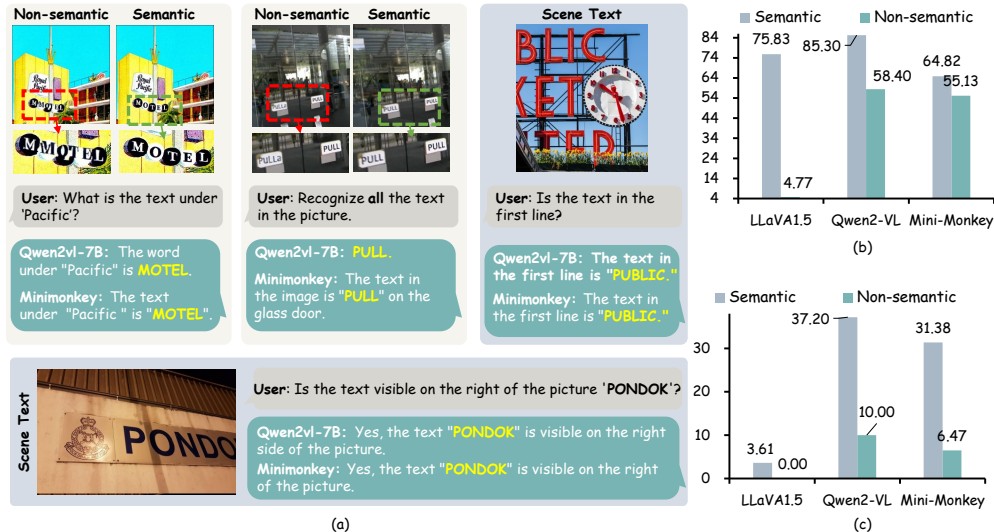

Figure 1: (a) LMMs hallucinate scene-text answers by relying on semantic priors rather than grounding in the actual visual content. For instance, when we edit "MOTEL" and "PULL" to "MMOTEL" and "PULLa", the models still answer the original ones. (b) and (c) illustrate the performance of LMMs on OCRBench and ICDAR 2015, with separate evaluations on semantic and non-semantic text samples.

To spot and understand scene texts, traditional approaches [9, 10, 11, 12] rely on multi-stage methods, separately addressing text detection, recognition, and language modeling, which limits their generalization ability in diverse real-world settings. As a general solution for vision-language tasks, Large Multimodal Models (LMMs) [13, 14, 15, 16, 17] have shown remarkable capabilities in image captioning and visual question answering by combining visual encoders with Large Language Models (LLMs). Motivated by this progress, researchers have begun adapting LMMs for OCR-related tasks, including document question answering [18, 19, 20, 21], GUI analysis agents [22, 23], and unified OCR frameworks [24].

However, whether LMMs can reliably address scene text spotting and understanding remains under-explored. In this work, we investigate this question through a "TextTrap" challenge. As illustrated in Fig. 1, LMMs such as Qwen2-VL [13] perform well when scene texts are semantically coherent. However, introducing subtle character-level perturbations that disrupt semantic meaning often leads these models to produce semantically plausible yet visually incorrect answers, a phenomenon we refer to as **semantic hallucination**. Further experiments on ICDAR 2015 [25] and OCRBench [26] provide solid evidence that LMMs frequently hallucinate scene text answers based on semantic priors rather than actual visual grounding.

Motivated by the intuition that semantic priors mainly originate from the LLM, we analyze the causes of hallucination from two perspectives. Inspired by prior observations [27, 28, 29] that different layers in LLMs capture different types of information, we further reveal that these layers exhibit varying tendencies to hallucinate, with certain intermediate layers showing a higher likelihood of correctly predicting ground-truth tokens. Building upon this insight, we further quantify and inspect the spatial distribution of attention maps within the LLM, and observe that layers allocating greater attention to ground-truth text regions are less prone to hallucination, **thereby suggesting a causal relationship between accurate attention allocation and the mitigation of semantic hallucination.**.

Based on these findings, we propose a semantic hallucination mitigation framework composed of two key components: **ZoomText**, which takes a "glimpse-refocus" steps to first localize contextual regions related to the scene text, and then refines its focus to estimate scene text regions. This coarse-to-fine grounding strategy eliminates the need for external model intervention. **Grounded Layer Correction (GLC)**: Given the anchor regions produced by ZoomText, GLC adaptively selects the transformer layer with the strongest scene text grounding and fuses its hidden state representations into the decoding process. This design helps mitigate hallucinations for non-semantic samples while preserving the semantics of meaningful ones. Notably, our method is training-free and can be

seamlessly integrated into existing LMMs to effectively mitigate semantic hallucination in scene text spotting and understanding.

Our main contributions are summarized as follows: 1) We identify the problem of semantic hallucination in LMMs when spotting and understanding scene text. We further investigate its underlying causes, revealing that attention drift across different layers within the LLM contributes significantly to hallucination. 2) We propose a training-free hallucination mitigation framework that can be seamlessly integrated into existing LMMs without requiring any architectural modifications. 3) We conduct extensive experiments on multiple benchmarks, demonstrating the effectiveness of our method. For example, when applied to the Mini-Monkey [30] and Qwen2.5-VL [13], our framework yields substantial accuracy gains on ST-VQA [5] and TextVQA [31]. Additionally, we introduce TextHalu-Bench, a new benchmark designed to evaluate semantic hallucination, where our framework consistently improves existing methods by approximately 4%.

## 2 Related Works

**Large Multimodal Models for OCR.** LMMs have demonstrated strong performance in general visual understanding tasks such as image captioning [13, 14, 15, 32, 33], visual question answering [16, 34, 17, 35, 36, 37, 38, 39, 40], and video understanding [41, 42, 43, 44, 45, 46]. However, the increasing demand for text-grounded visual reasoning has revealed its limitations in accurate OCR. Recent works have proposed OCR-specific enhancements for LMMs, which can be broadly categorized into three strategies. (1) *Resolution-aware processing*: UReader introduces shape-adaptive cropping [18], while Monkey [19] and TextMonkey [20] adopt patch-wise division to better handle high-resolution text regions. Ocean-OCR [47] further utilizes a native-resolution ViT to support variable input sizes. (2) *Token compression and layout encoding*: mPLUG-DocOwl [21] and TextHawk2 [48] reduce visual token redundancy while preserving spatial structure. Vary [49] introduces a SAM-style [50] visual vocabulary tailored for document and chart understanding. (3) *Redesigned OCR paradigms*: GOT-OCR [24] proposes a new task formulation and architecture specifically optimized for OCR scenarios. Despite these advances, current models still rely heavily on semantic priors and often fail when the input contains visually plausible but meaningless words. This indicates a lack of text grounding. Our work investigates this failure mode and proposes an attention-based inter-layer fusion mechanism to enhance robustness in text-level reasoning.

**Hallucination in Large Multimodal Models.** Hallucination in LMMs refers to the generation of outputs that are not grounded in the visual input, often leading to content that is irrelevant or factually incorrect. Prior work has systematically explored hallucination along several dimensions, including object hallucination [51, 52, 53, 54, 55], knowledge hallucination [56, 54, 57], relational misinterpretation [58, 59, 57, 60], attribute hallucination [58, 57, 60, 55], and hallucination induced by spurious visual patterns [61, 62]. Recent studies have also revealed inconsistencies in model responses across question types [63, 64, 65]. To mitigate hallucination, various strategies have been proposed, including self-correction decoding [66, 67, 68], contrastive decoding [69, 70], and adversarial training [64, 56]. However, most of these studies focus on object- or fact-centric hallucinations, while OCR-specific hallucinations remain underexplored. In this work, we identify a novel form of *semantic hallucination in scene text spotting*: LMMs can accurately recognize semantically meaningful words, yet fail when those words are replaced with syntactically valid but semantically meaningless tokens. This behavior indicates that models rely heavily on semantic priors rather than truly grounding their predictions in visual evidence.

## 3 Methods

In this section, we first provide background on the generation paradigm of LMMs and analyze the underlying causes of semantic hallucination in scene text spotting and understanding. These analyses reveal that semantic hallucination is closely tied to attention drift within LLMs, where attention deviates from ground-truth text regions. Building upon these insights, we introduce our training-free hallucination mitigation framework.

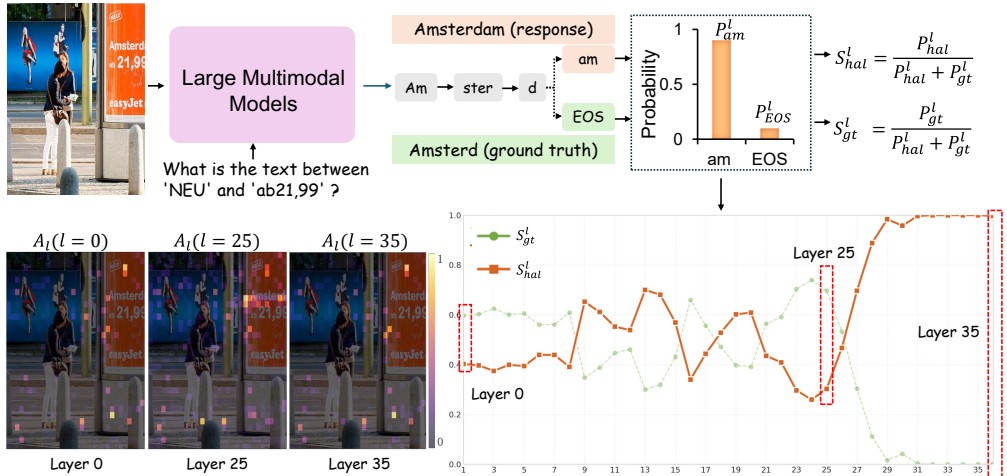

Figure 2: Visualization of the hallucination-analysis pipeline. For each input image, we (1) identify hallucinated text tokens and compute their layer-wise hallucination-tendency scores, (2) calculate the ratio of the ground-truth text score to the hallucinated text score for each layer, and (3) overlay these normalized ratios onto the corresponding attention maps. We observe that layers with a lower propensity to hallucinate concentrate their attention more strongly on the text regions.

## 3.1 Preliminaries of LMMs Generation

Most current LMMs adopt the minimalist architecture of LLaVA [35], which comprises a visual encoder, a vision-language projector, and an LLM. Given an input image, the visual encoder extracts a sequence of visual tokens $V = \{v_1, v_2, \dots, v_n\}$, where $n$ denotes the number of output patches. Similarly, the text input is tokenized into a sequence of text tokens $T = \{t_1, t_2, \dots, t_m\}$. These two token sequences are concatenated as $X = \texttt{concat}(V, T)$ and fed into the LLM, parameterized by $\theta$, for auto-regressive generation. At each decoding step $i$, the model predicts the probability distribution over the next token $y_i$ in an auto-regressive manner:

$$p(y_i \mid V, T, y_{<i}) = \text{softmax}\left(\text{logit}_\theta(y_i \mid V, T, y_{<i})\right) \tag{1}$$

To generate the final output, decoding strategies such as greedy decoding or beam search are employed to select the next token. The predicted token $y_i$ is then appended to the previous input sequence, and the process is repeated until a stop condition is met.

## 3.2 Investigating the Mystery of Semantic Hallucination

LMMs are pretrained on large-scale corpora primarily composed of semantically coherent texts, which may impose strong semantic priors on the model. In scene text spotting and understanding, such priors may cause the model to incorrectly interpret visually meaningless or random character patterns as meaningful words. To gain deeper insights into how these hallucinations arise within the model, we focus on the internal processing of the LMM. Prior work shows that different layers capture different types of information [68]. Building on this, we hypothesize that different layers of the LLM may exhibit varying tendencies to produce semantic hallucinations. To validate this hypothesis, we design an analysis pipeline consisting of two steps:

- **Hallucinated Token Extraction.** For each generated output, we tokenize both the generated answer and the ground-truth answer using the LMM's predefined text tokenizer. We then compare the two token sequences and identify the first token in the generated sequence that diverges from the ground-truth as a hallucinated token.

- **Hallucination Tendency Scoring.** We compute the hallucination tendency score at each layer $\ell$ by comparing the output probabilities of the hallucinated token and its ground-truth counterpart. Specifically, at each decoding step $t$, the model computes a probability distribution over the entire

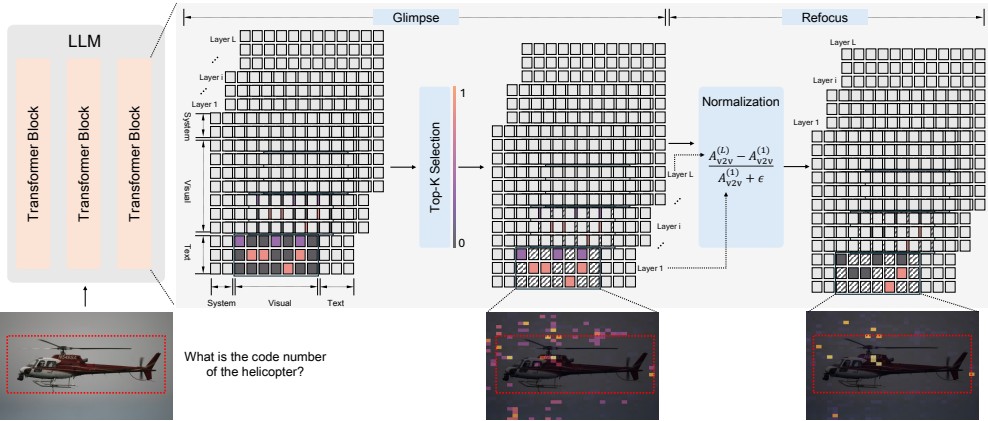

Figure 3: Visualization of the ZoomText process and examples.

vocabulary based on the prefix $x_{<t}$. From this distribution, we extract the probabilities assigned to both $y_{\text{hal}}$ and $y_{\text{gt}}$ as candidate tokens.

$$P_{\text{hal}}^{\ell} = \text{softmax}(\boldsymbol{W}_{\text{out}}\boldsymbol{h}_{\text{hal}}^{\ell} + \boldsymbol{b})_{y_{\text{hal}}}, \qquad P_{\text{gt}}^{\ell} = \text{softmax}(\boldsymbol{W}_{\text{out}}\boldsymbol{h}_{\text{gt}}^{\ell} + \boldsymbol{b})_{y_{\text{gt}}} \qquad (2)$$

where $\boldsymbol{W}_{\text{out}}$ and $\boldsymbol{b}$ are the parameters of the output head. The hallucination score $S_{\text{hal}}^{\ell}$ is then calculated by $P_{\text{hal}}^{\ell}/(P_{\text{hal}}^{\ell} + P_{\text{gt}}^{\ell})$. A higher $S_{\text{hal}}^{\ell}$ indicates that the model is more likely to favor the hallucinated output over the correct one at layer $\ell$.

As shown in Fig. 2, different transformer layers within the LMM exhibit varying tendencies toward semantic hallucination, with more examples provided in the Supplementary Material.

Based on this observation, we aim to further investigate the underlying mechanisms driving these differences, particularly focusing on the visual grounding behavior of different layers (i.e., how they attend to relevant scene text regions). This leads us to pose a key question: *Is there a relationship between a transformer layer's visual grounding ability (specifically, its attention to scene text regions) and its tendency to produce semantic hallucinations?*

To answer this question, we propose a quantitative measure of visual grounding for each layer, termed the *Text-region Attention Score* ($A_\ell$). This score evaluates how much attention a transformer layer allocates to ground-truth text regions, which is calculated as:

$$A_\ell = \frac{\sum_{i\in\mathcal{I}}\sum_{j\in\mathcal{T}}\alpha_{i,j}^{\ell}}{\sum_{i\in\mathcal{I}}\sum_{j\in\mathcal{I}}\alpha_{i,j}^{\ell}} \qquad (3)$$

where $\mathcal{I}$ denotes the set of all image tokens, and $\mathcal{T}\subset\mathcal{I}$ represents those image tokens located within the provided ground-truth text bounding boxes. $\alpha_{i,j}^{\ell}$ is the self-attention weight from the $i$-th image token to the $j$-th image token at layer $\ell$. Higher values of $A_\ell$ reflect an increased allocation of attention score to the correct text regions, indicative of more robust visual grounding at layer $\ell$.

Based on Qwen2.5-VL [13] and Mini-Monkey [30], we evaluate our method on OCRBench [26], ST-VQA [5], and TextVQA [31], and analyze the Spearman correlation [71] between layer-wise hallucination tendency scores and their corresponding text-region attention scores. We observe a strong negative correlation across all datasets, indicating that layers with lower attention to ground-truth text regions are more susceptible to semantic hallucination. Additional experimental details are provided in Appendix E.

### 3.3 Toward Semantic Hallucination Mitigation

Building on this key observation about semantic hallucination, we aim to leverage the connection between visual grounding ability and hallucination tendency to design an effective mitigation strategy. This naturally raises two questions: 1) How can we estimate scene text regions without relying on

additional modules? 2) Once we identify the layer with the strongest scene text grounding, how can we guide the decoding process using this information to reduce hallucinations?

**ZoomText.** Unlike naturally salient objects, scene text is often difficult to localize, especially in the absence of external text detectors. To address this challenge, we propose a *glimpse-refocus* strategy for estimating scene text regions. We begin by observing that scene text frequently appears on semantically meaningful backgrounds, such as signs, posters, or product packaging, which naturally attract model attention during question answering [68]. Leveraging this intuition, we perform a *glimpse* step that identifies text-related regions by computing the query-to-image cross-attention, which measures how much each image token contributes to the query understanding. The highlighted attention regions serve as a coarse estimation of potential text positions. Specifically, we extract the softmax-normalized cross-attention from the query tokens to all image tokens at the final layer of the LLM, resulting in $A_{\text{q2v}} \in \mathbb{R}^{H \times Q \times N}$, where $H$ is the number of attention heads, $Q$ is the number of query tokens, and $N$ is the number of image tokens. We average across heads and query tokens to obtain a global image attention map:

$$A_{\text{text}} = \frac{1}{HQ} \sum_{h=1}^{H} \sum_{q=1}^{Q} A_{\text{q2v}}^{(h,q)} \in \mathbb{R}^{N}. \tag{4}$$

We then apply thresholding to select the top-$K$ image tokens as coarse text region candidates.

However, not all high-response tokens are truly relevant to the query, as LLMs often utilize certain tokens as "registers" to aggregate global context across the image. To mitigate this bias toward irrelevant regions, we introduce a *Refocus* step that filters out spurious activations. This step is based on the hypothesis that background or non-semantic tokens exhibit relatively stable attention patterns across layers, as they do not actively participate in the visual reasoning process. Accordingly, we compute a normalized attention shift score among the top-$K$ candidate tokens identified in the *Glimpse* stage, which quantifies how much each token's importance evolves throughout the forward pass. Let $\mathcal{S} = \{s_1, \ldots, s_K\}$ denote the set of top-$K$ image token indices selected from $A_{\text{text}}$. We extract the self-attention submatrices $A_{\text{v2v}}^{(1)}$ and $A_{\text{v2v}}^{(L)} \in \mathbb{R}^{K \times K}$ from the first and last transformer layers, and then compute a normalized attention shift score as:

$$A_{\text{text}}^{\text{normalized}} = \frac{A_{\text{v2v}}^{(L)} - A_{\text{v2v}}^{(1)}}{A_{\text{v2v}}^{(1)} + \epsilon} \tag{5}$$

where $\epsilon$ is a small constant for numerical stability. As shown in Fig. 3, most noisy tokens are effectively filtered out, leaving accurate text regions that can be used to guide the decoding process.

**Grounded Layer Correction.** After identifying grounded text regions, we select the most visually grounded transformer layer $\ell^\star = \arg\max_\ell A_\ell$, and propose three strategies to correct the decoding process. All strategies operate on the final-layer hidden states $\boldsymbol{H}^{(L)}$ before decoding, producing revised representations $\hat{\boldsymbol{H}}$ that integrate information from the grounded layer $\boldsymbol{H}^{(\ell^\star)}$. Specifically, we either: (1) replace all hidden states with the grounded ones (Replacement), (2) apply a weighted fusion with a factor $w$ (Fusion), or (3) selectively replace tokens with high grounding scores based on the refined attention map $\mathcal{S}$ (Selective Replacement).

$$\hat{\boldsymbol{H}}_i = \begin{cases} \boldsymbol{H}_i^{(\ell^\star)} & \text{(Replacement)} \\ (1 - w) \cdot \boldsymbol{H}_i^{(L)} + w \cdot \boldsymbol{H}_i^{(\ell^\star)} & \text{(Fusion)} \\ \boldsymbol{H}_i^{(L)}, \text{ if } i \notin \mathcal{S}; \quad \boldsymbol{H}_i^{(\ell^\star)}, \text{ if } i \in \mathcal{S} & \text{(Selective Replacement)} \end{cases} \tag{6}$$

Empirical results (Sec. 5.3) show that among the three strategies, Fusion achieves the best balance between hallucination mitigation and semantic preservation. Accordingly, we adopt Fusion as our default decoding strategy.

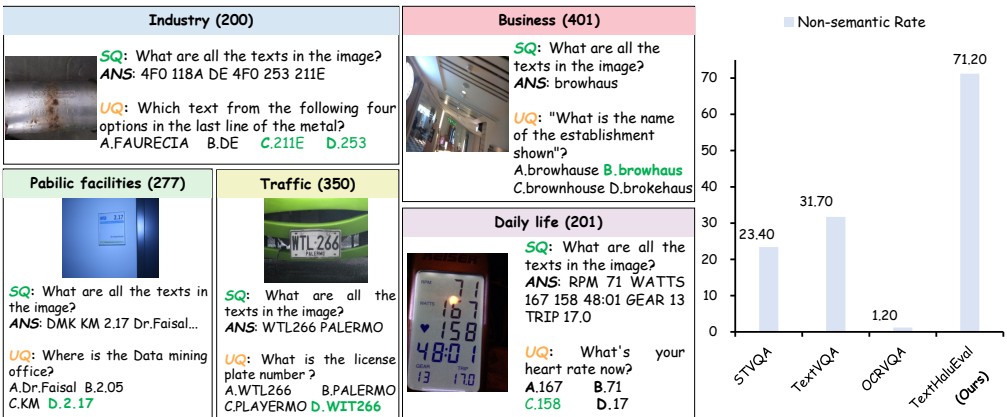

Figure 4: (a) Examples of TextHalu-Bench. (b) Comparison of non-semantic answers ratios between existing scene text benchmarks and TextHalu-Bench. SQ, UQ, and ANS represent spotting, understanding questions, and answers, respectively.

## 4 TextHalu-Bench

Previous scene-text benchmarks such as ST-VQA [5] and TextVQA [31] have notable limitations: their test sets are dominated by semantically meaningful and visually clear samples, as shown in Fig. 4. This may overestimate the visual grounding ability of LMMs, as models can often rely on language priors rather than true visual perception to answer correctly.

To address these limitations, we introduce **TextHalu-Bench**, a new benchmark comprising **1,740** carefully curated samples by select collected from diverse public datasets, including ICDAR 2013 [72], ICDAR 2015 [25], ICDAR 2019 [73], CCPD [74], MSRA-TD500 [75] ,RoadText [76] and MPSC [77]. The curation process specifically targets instances containing non-semantic text elements, such as isolated numbers, incomplete words, and rare or out-of-vocabulary tokens.

Our benchmark covers **five representative scenario categories**—*Business*, *Industry*, *Transportation*, *Public Facilities*, and *Daily Life*—with a balanced distribution and emphasis on visually challenging cases (e.g., occlusions, low-contrast text, unconventional fonts). It features two subtasks: **Spotting**, which requires models to extract text directly from images, and **Understanding**, which evaluates whether models can semantically ground the recognized text. The detailed data construction pipeline is provided in the Appendix C.

## 5 Experiments

### 5.1 Experimental Setup

**Baselines.** To validate the effectiveness of our proposed semantic hallucination mitigation framework, we integrate it into three contemporary open-source LMMs with diverse LLM backbones: **Mini-Monkey** [30], **Qwen2.5-VL** [78], and **LLaVA-NeXT** [37]. For all three models, we follow the default configurations provided in their official implementations to ensure a fair comparison. In addition, we evaluate our method alongside 10 representative LMMs, including both open-source and proprietary models, across multiple public benchmarks.

**Benchmarks.** In addition to our proposed TextHalu-Bench, we evaluate our method on six public benchmarks encompassing scene text spotting and understanding. ST-VQA [5] and TextVQA [31] focus on real-world images containing scene text, requiring models to understand and reason over both visual and textual information. AI2D [79] centers on scientific diagrams, emphasizing structured reasoning and domain-specific knowledge. OCR-VQA [80] involves book covers and challenges models to incorporate OCR-derived content into question answering. SEED-Bench [81] offers a broad suite of vision-language tasks; we evaluate its Text Understanding subset, which tests general VQA and grounding capabilities. Finally, GOT [24] contains 400 natural images with multilingual scene text; we use its scene text subset and report character-level F1 scores.

Table 1: Experimental results on TextHalu-Bench and mainstream scene text spotting and understanding benchmarks. We report the performance on STVQA and GOT by using their official weight.

| Method | LLM | TextHalu-Bench | STVQA$_{Test}$ | TextVQA$_{Val}$ | GOT$_{Scene}$ | OCRVQA$_{CORE}$ | SEEDBench$_{Text}$ | AI2D |
|---|---|---|---|---|---|---|---|---|
| **Proprietary Models** | | | | | | | | |
| Gemini1.5-Pro [82] | - | 43.2 | - | 61.6 | - | 18.5 | 76 | 79.1 |
| GPT-4o [83] | - | 45.3 | - | 71.0 | - | 18.7 | 70.2 | 85.9 |
| **Open-source MLLMs** | | | | | | | | |
| LLaVA1.5 [35] | Vicuna-7B | 21.4 | 51.9 | 46.0 | 38.8 | 60.6 | 36.9 | 55.5 |
| mPLUG-Owl2 [84] | LLaMA-7B | 24.3 | 49.8 | 56.4 | 29.8 | 65.2 | 32.1 | 55.7 |
| Molmo-D [85] | Qwen2-7B | 24.7 | 62.3 | 67.5 | 42.3 | 15.9 | 77.4 | 81.0 |
| PixtralB [86] | Nemo-12B | 32.8 | 52.9 | 64.3 | 35.4 | 64.7 | 47.6 | 79.0 |
| Monkey [87] | Qwen-7B | 34.2 | 54.7 | 67.6 | 45.7 | 67.0 | 56.0 | 62.5 |
| LLaVA-OV [88] | Qwen-2.7B | 21.4 | 51.9 | 78.5 | 43.9 | 64.7 | 61.9 | 82.8 |
| Ovis1.6 [89] | Llama-3.2-3B | 38.4 | 72.6 | 78.2 | 25.2 | 71.2 | 52.4 | 84.4 |
| InternVL2.5 [90] | InternLM2.5-7B | 42.0 | 75.4 | 79.0 | 90.0 | 31.0 | 77.1 | 84.2 |
| LLaVA-NeXT [38] | Llama-3-8B | 27.9 | 65.1 | 65.3 | 41.9 | 60.7 | 50.0 | 72.8 |
| **LLaVA-NeXT + Ours** | Llama-3-8B | **28.5** (+0.6) | **65.2** (+0.1) | **65.5** (+0.2) | **42.0** (+0.1) | **61.5** (+0.8) | **51.2** (+1.2) | **73.0** (+0.2) |
| Mini-Monkey [91] | InternLM2-1.8B | 46.5 | 66.7 | 74.1 | 88.8 | 39.7 | 83.3 | **74.8**) |
| **Mini-Monkey + Ours** | InternLM2-1.8B | **50.6** (+4.1) | **70.6** (+3.9) | **75.0** (+0.9) | **89.2** (+0.4) | **39.9** (+0.2) | **84.5** (+1.2) | 74.7 (–0.1) |
| Qwen2.5-VL [78] | Qwen2.5-3B | 48.3 | 67.3 | 79.1 | 85.2 | 70.2 | 66.7 | 78.1 |
| **Qwen2.5-VL + Ours** | Qwen2.5-3B | **53.8** (+5.5) | **67.6** (+0.3) | **80.3** (+1.2) | **86.0** (+0.8) | **70.5** (+0.3) | **70.2** (+3.5) | **78.3** (+0.2) |

**Implementation Details.** Our method is a training-free and test-time adaptive plug-in module. In ZoomText, we set the number of top image tokens $K$ to 128. In Grounded Layer Correction, we adopt the Fusion strategy and set the fusion factor $w$ to 0.1. All experiments are conducted on a single NVIDIA A800-80G GPU during inference. Importantly, our algorithm introduces no additional modules or trainable parameters. Test-time efficiency analysis is provided in Appendix E.

## 5.2 Experiment results

We conduct extensive experiments on the seven benchmarks, as demonstrated in Tab. 1, in which we derive three primary conclusions.

**Semantic hallucination remains a significant challenge for existing LMMs.** On our proposed TextHalu-Bench, even the best-performing proprietary model, GPT-4o, achieves only a 45.3 F1 score, while most open-source models perform considerably worse, far below human performance (96.8). This difficulty arises from two key aspects. First, compared to document-based OCR tasks, scene text spotting and understanding are inherently more challenging due to the presence of complex visual distractors and highly diverse text styles. Second, non-semantic texts require accurate visual grounding rather than reliance on semantic priors, an area where many LMMs still suffer from severe hallucinations. These findings highlight the urgency of addressing semantic hallucination and underscore the importance of TextHalu-Bench, which incorporates diverse non-semantic texts to robustly evaluate and analyze the hallucination behavior of LMMs.

**Effectiveness of the proposed hallucination mitigation method.** We integrate our method into three LMMs with different underlying LLM architectures. Mini-Monkey and Qwen2.5-VL achieve **4.1%** and **5.5%** improvements in F1 score respectively, indicating that our method effectively helps models remain faithfully grounded on visual cues for scene text spotting and understanding. In contrast, LLaVA-Next shows only a marginal improvement of 0.6%, which we attribute to its limited OCR-related capabilities. These results suggest that our method can bring greater benefits when applied to models with stronger scene text perception abilities.

**Generalization to other benchmarks.** Beyond TextHalu-Bench, our method demonstrates promising results on a range of public vision-language benchmarks centered on scene text understanding and spotting. All baseline models show consistent improvements when integrated with our framework. Notably, Mini-Monkey achieves an accuracy gain of approximately 4% on ST-VQA, while Qwen2.5-VL improves by around 3% on SEED-Bench. These results suggest that our hallucination mitigation approach serves as a generalizable solution, effectively enhancing visual grounding without compromising the original recognition capabilities on semantically valid samples.

Table 2: Comparison of different hallucination mitigation methods. "Adv." means adversarial training method, and "CoT" means Chain-of-Thought testing strategy.

| Methods | TextHalu-Bench | STVQA$_{Test}$ | TextVQA$_{Val}$ | GOT$_{Scene}$ | OCRVQA$_{CORE}$ | SEEDBench$_{Text}$ | AI2D |
|---------|----------------|----------------|-----------------|---------------|-----------------|-------------------|------|
| Baseline | 46.5 | 66.7 | 74.1 | 88.8 | 39.7 | 83.3 | 74.8 |
| Adv. | 47.5 (+1.0) | 66.8 (+0.1) | 73.7 (–0.4) | 89.1 (+0.3) | **39.9** (+0.2) | 83.3 (+0.0) | 74.5 (–0.3) |
| CoT | 46.8 (+0.3) | 68.2 (+1.5) | 75.2 (+1.1) | **89.2** (+0.4) | 39.7 (+0.0) | 83.3 (+0.0) | **74.9** (+0.1) |
| Ours | **50.6** (+4.1) | **70.6** (+3.9) | **75.0** (+0.9) | **89.2** (+0.4) | **39.9** (+0.2) | **84.5** (+1.2) | 74.7 (–0.1) |

Table 3: Ablations about the effectiveness of ZoomText.

| Methods | TextHalu-Bench | STVQA$_{Test}$ | TextVQA$_{Val}$ | GOT$_{Scene}$ | OCRVQA$_{CORE}$ | SEEDBench$_{Text}$ | AI2D |
|---------|----------------|----------------|-----------------|---------------|-----------------|-------------------|------|
| Baseline | 46.5 | 66.7 | 74.1 | 88.8 | 39.7 | 83.3 | **74.8** |
| with text detector | 50.4 (+3.9) | **70.8** (+4.1) | **75.2** (+1.1) | 89.0 (+0.2) | **39.9** (+0.2) | 83.3 (+0.0) | 74.7 (-0.1) |
| w/o Glimpse | 50.2 (+3.7) | 70.2 (+3.5) | 75.0 (+0.9) | 88.7 (-0.1) | 39.8 (+0.1) | **84.5** (+1.2) | **74.8** (+0.0) |
| w/o Refocus | 49.8 (+3.3) | 69.5 (+2.8) | 74.9 (+0.8) | 88.7 (-0.1) | 39.7 (+0.0) | 83.3 (+0.0) | **74.8** (+0.0) |
| **Ours** | **50.6** (+4.1) | 70.6 (+3.9) | 75.0 (+0.9) | **89.2** (+0.4) | **39.9** (+0.2) | **84.5** (+1.2) | 74.7 (-0.1) |

## 5.3 Ablation Experiment

We conduct extensive ablation studies to evaluate the robustness and generalization capability of our proposed semantic hallucination mitigation method. Mini-Monkey is used as the primary baseline, and results on additional models are provided in Appendix F.

**Comparison with other hallucination mitigation methods.** As most existing hallucination mitigation techniques, such as contrastive decoding and self-correcting decoding, are not directly applicable to our setting, we design two tailored baselines for comparison. (1) *Training-based adversarial training:* Following [64], we construct leading question–answer pairs to augment the training set with adversarial examples, and retrain the LMM using this data. (2) *Training-free Chain-of-Thought (CoT) prompting:* We apply CoT prompts to guide the model to first attend to text regions before generating answers. Further implementation details are provided in the Appendix F. As shown in Tab. 2, adversarial training yields only marginal improvements on TextHalu-Bench. While the CoT strategy enhances attention to text regions and improves performance on general scene text tasks, it fails to fundamentally address semantic hallucination.

**Effectiveness of ZoomText.** Our proposed ZoomText module estimates potential scene text regions without relying on external text detectors. We validate its effectiveness in two ways. First, we compare ZoomText with a baseline that incorporates accurate region proposals obtained from an off-the-shelf pretrained text detector [92]. Second, we ablate ZoomText's two key components, *Glimpse* and *ReFocus*, to evaluate their individual contributions. As shown in Tab. 3, ZoomText achieves performance comparable to models equipped with external detectors, demonstrating its standalone effectiveness. Moreover, we observe that both Glimpse and ReFocus contribute significantly to performance, highlighting the importance of coarse-to-fine region localization.

**Ablation on Grounded Layer Correction (GLC).** To demonstrate the effectiveness of GLC, we first ablate the impact of our layer selection strategy. Specifically, we randomly select a layer from the early, middle, and late stages of the LLM and apply the same Fusion strategy. As shown in Fig. 5, intermediate layers can indeed help mitigate hallucination. However, they may also overwrite valid semantic knowledge, as reflected by performance drops on general VQA benchmarks such as ST-VQA. In contrast, our method adaptively selects the layer with the strongest scene text grounding, leading to reduced hallucination on non-semantic samples while preserving the semantic integrity of meaningful ones. Furthermore, we evaluate the three correction strategies introduced in Sec. 3.3: *Replacement*, *Selective Replacement*, and *Fusion*. For fair comparison, all methods operate on the same grounded layer identified by our selection strategy. Naive Replacement performs poorly across all benchmarks, likely due to a significant domain gap between training-time representations and directly injected hidden states. In contrast, both Selective Replacement and Fusion effectively reduce hallucinations. However, similar to the trend observed in layer selection, Selective Replacement substantially degrades performance on general scene text understanding tasks. We attribute this to its aggressive overwriting of final-layer hidden states, which may disrupt the learned alignment between

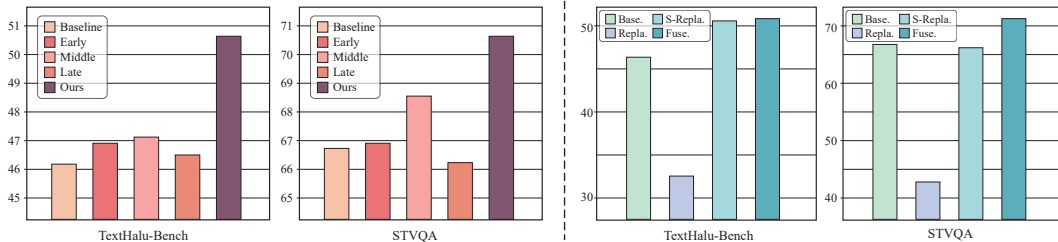

Figure 5: Ablation on the Grounded Layer Correction. **(Left)** Different layer selection method. **(Right)** Different correction strategy. "Base": Baseline; "Repla.": Replacement; "S-Repla.": Selective Replacement; "Fuse": Fusion.

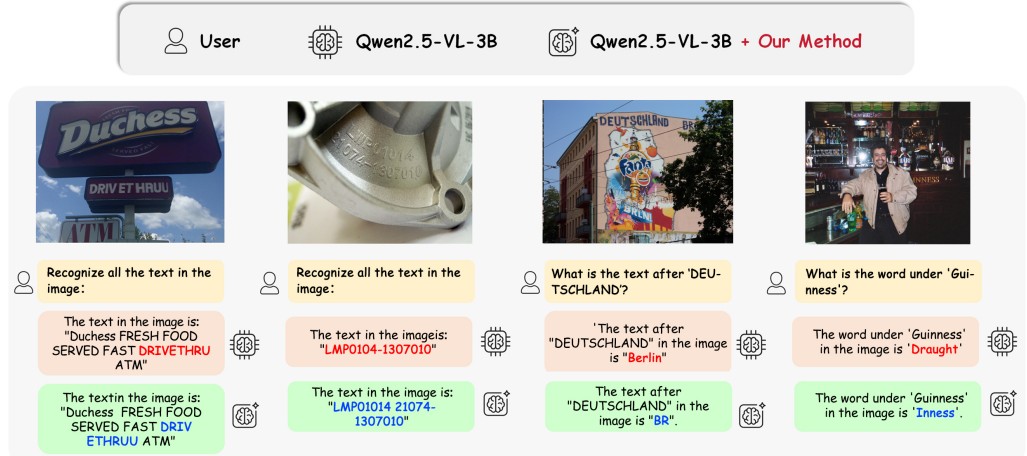

Figure 6: Visualization of our proposed methods.

visual text and multimodal context. In light of these results, we adopt *Fusion*, a weight-controlled integration, as our default strategy. Details on fusion weight selection are provided in Appendix F.

## 6 Visualization

We provide visualization results in Fig. 6, including synthetic images generated with [93] and challenging samples from our proposed TextHalu-Bench. As shown, base models such as Qwen2.5-VL demonstrate limited ability to accurately spot texts without reasonable semantics, such as sign names and numbers. Moreover, these models are prone to hallucinating semantic words (e.g., "Berlin") that do not appear in the images. In contrast, our proposed method enables models to respond to questions grounded in the target regions of images, thereby improving the reliability of scene text spotting and understanding.

## 7 Conclusion

In this work, we identify the problem of semantic hallucination in Large Multimodal Models, where models often produce semantically plausible but visually incorrect answers when spotting and understanding scene text. We analyze its underlying causes and establish a strong correlation between accurate intra-layer attention allocation and the reduction of semantic hallucination. Building on this insight, we propose a training-free hallucination mitigation framework comprising two key components. First, *ZoomText* adopts a coarse-to-fine strategy to estimate scene text regions without relying on external detectors. Second, *Grounded Layer Correction* leverages the hidden states from the most visually grounded layer to guide the decoding process. Furthermore, we introduce *TextHalu-Bench*, a benchmark designed to robustly evaluate scene text spotting and understanding in the presence of non-semantic text. Extensive experiments demonstrate the effectiveness and generalizability of our approach across multiple LMMs and benchmarks.

## Acknowledgements

This work was supported by the Fundamental Research Funds for the Central Universities (Grant No. 3262025T82) and partly supported by the EU Horizon projects ELIAS (No. 101120237) and ELLIOT (No. 101214398).

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

# A  Overview of Appendix

# B  Limitations and Broader Impact

**Limitations.** While our method shows promising performance on scene text spotting and understanding, it still has two key limitations. First, it requires token selection and attention map computation during the prefilling stage, which introduces additional inference time and computational overhead. Second, the effectiveness of our method heavily relies on the underlying OCR perception ability of the base model. As a result, it performs suboptimally when applied to LMMs with weak scene text understanding capabilities.

We propose a training-free semantic hallucination mitigation framework with broad impacts across multiple domains. For the OCR community, our method facilitates the adaptation of LMMs to text-intensive tasks, potentially benefiting downstream applications including document understanding, autonomous driving, assistive technologies, and low-level text processing techniques [94, 95] such as editing and generation. Beyond OCR, our findings and mitigation strategy provide valuable insights for developing more reliable and hallucination-resilient multimodal large models. Importantly, this framework is generalizable and can be extended to other domains where visual-semantic alignment is critical, such as remote sensing image interpretation [96, 97] and medical image analysis [98], where hallucination mitigation is equally crucial for accurate and trustworthy predictions.

# C  Details of TextHalu-Bench

**Dataset Collection Process.** To promote coverage and diversity, we carefully curated samples across five representative scenario types: *Business*, *Industry*, *Transportation*, *Public Facilities*, and *Daily Life*. These categories were selected based on their prevalence in real-world OCR applications and their variance in textual layout, typeface complexity, and visual background noise. In addition, during sample construction, we emphasized the inclusion of challenging edge cases, such as low-contrast text, occlusions, unconventional fonts, or partial visibility, to stress-test the visual grounding ability of MLLMs and better surface hallucination tendencies.

**Scene Text Spotting Task Definition.** Given an image, the model is required to extract all visible textual content from the scene. The task is treated as a word-level prediction problem and its output is compared to the ground-truth words using case-insensitive exact match. Spotting examples include questions such as "What is the texts in the image? Answer the question in only words you recognize."

**Scene Text Understanding Task Definition.** To measure the higher-level comprehension ability, we adopt a *multiple-choice format*, with at least one correct answer and at most three well-crafted distractors. These distractors are designed with the following strategies:

- **Glyph-based distractors**: visually similar characters (e.g., 'O' vs. '0', 'l' vs. '1')
- **Semantic distractors**: misleading but contextually related words (e.g., "apple" vs. "apole")
- **Context-based distractors**: co-occurring or spatially nearby words within the same image

Understanding task examples include questions such as "What is the texts on the boat? A.aa B.bb C.cc D.dd "

**Metric.** To quantitatively measure hallucination behavior, we report the average F1 score across both subtasks as our evaluation metric which captures both the model's accuracy in extracting visible text and its ability to semantically interpret visual information, distinguishing genuine visual understanding from language-prior hallucination.

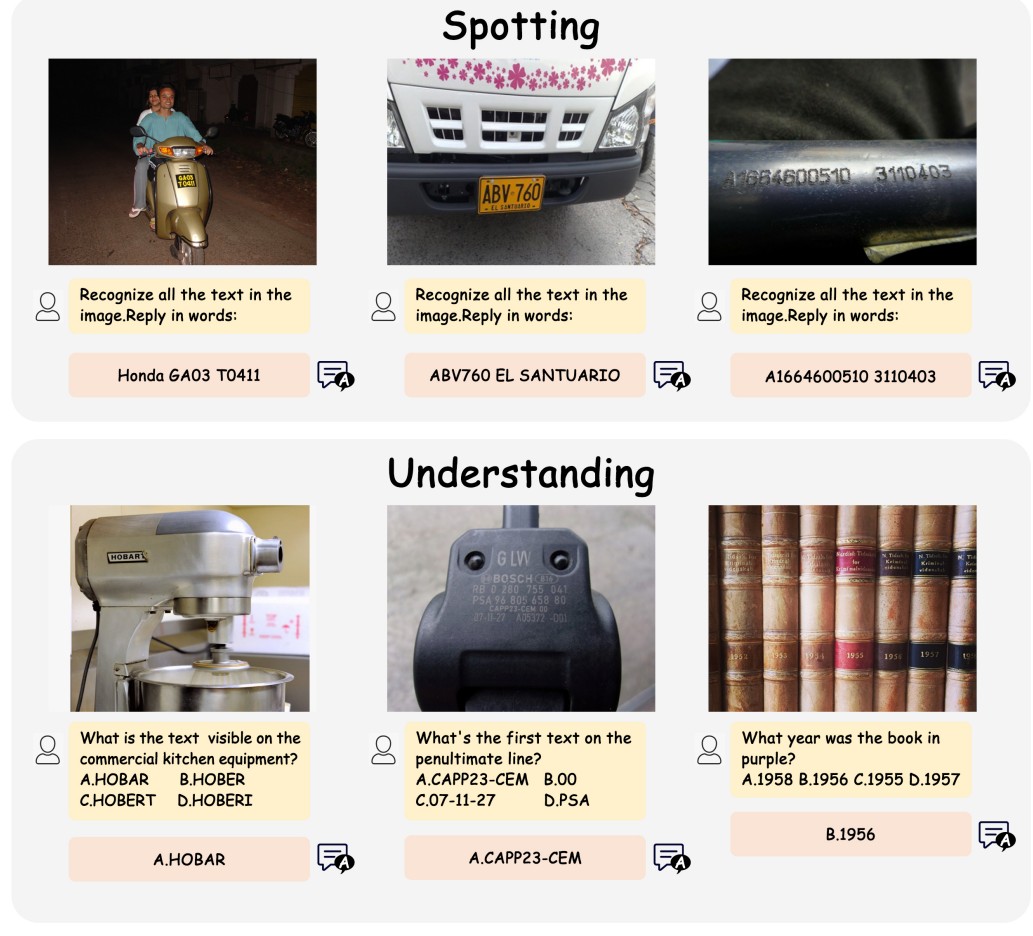

Figure 7: Visualization of TextHalu-Bench.

Table 4: Spearman Correlation between Hallucination tendency score and scene text region attention score, with performance on STVQA and TextVQA.

| Model | Benchmarks | | |
|---|---|---|---|
| | OCRBench | STVQA | TextVQA |
| Mini-Monkey | -0.72 | -0.78 | -0.74 |
| Qwen2.5-VL | -0.68 | -0.80 | -0.76 |

**Visualizations.** We provide some qualitative cases of our benchmark in Fig. 7.

## D   Experimental Settings

Our method is training-free and thus does not require any additional fine-tuning or parameter updates. All models are evaluated under their official default configurations without modification. For evaluation, we test on TextHalu-Bench, ST-VQA, and GOT using our own implementation to ensure consistent handling of OCR and visual inputs. For other benchmarks, we utilize the **VLMEvalKit** [99] toolkit, and follow the original leaderboard results published by each model for a fair comparison. All experiments are conducted on a server equipped with 4 × NVIDIA A800 GPUs.

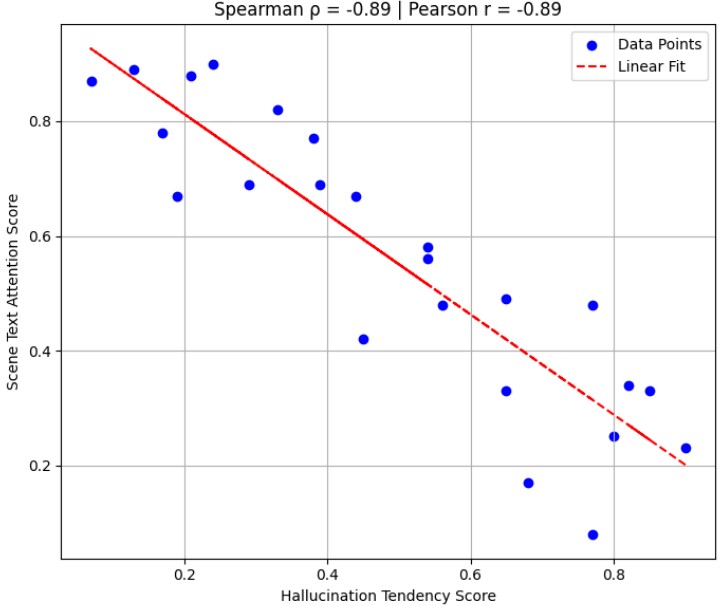

Figure 8: An example of the correlation (Spearman and Pearson coefficient) between hallucination tendency score and scene text region attention score.

Table 5: Generalization performance of our method on other domains.

| Method | SEEDBench | RealWorldQA | MathVista | POPE | MME-P | MME-VideoOCR | A-OKVQA |
|---|---|---|---|---|---|---|---|
| Qwen2.5VL | 74 | 65.5 | 61.2 | 85.9 | 1567.7 | 59.2 | 85.2 |
| **Qwen2.5VL + Ours** | **74.1** | **65.8** | **61.4** | **86.7** | **1572.2** | **60.8** | **85.6** |

# E  More Experimental Results

**Correlation between Hallucination and Attention Distribution in LLMs.** Leveraging our automatic hallucinated token identification mechanism, we compute hallucination tendency scores and corresponding scene text region attention scores across all transformer layers for each hallucinated sample. As shown in Tab. 4, Spearman correlation analysis reveals a strong negative correlation, indicating that stronger attention to scene text regions is associated with reduced semantic hallucination. A layer-wise visualization for a representative sample is shown in Fig. 8, where each point corresponds to a transformer layer with its hallucination score (x-axis) and scene text attention score (y-axis).

**Generalization to other domains.** To assess the generalization ability of our method beyond the scene text domain, we apply it to four diverse vision-language benchmarks: **SEED-Bench** consists of 19K multiple-choice questions with accurate human annotations, covering 12 evaluation dimensions across both image and video modalities; **RealWorldQA** [100] evaluates real-world spatial understanding in physical environments, contributed by XAI; **MathVista** [101] is a challenging benchmark requiring visual mathematical reasoning over charts, diagrams, and textual math problems; **POPE** [51] focuses on object hallucination, comprising three evaluation tracks: random, popular, and adversarial hallucination. **MME** [102] is a large-scale comprehensive multimodal benchmarks toward the perception and reasoning ability of LMMs. **MME-VideoOCR** [103] focuses on the multi level ability on the video text understanding. A-OKVQA [104] is a challenging benchmark that requires commonsense and world knowledge to answer.

As shown in Tab. 5, our method consistently improves performance across all benchmarks—for instance, achieving **+0.3** accuracy gain on RealWorldQA and **+0.8** on POPE. These results suggest that our approach not only enhances scene text understanding but also generalizes well to broader multimodal reasoning tasks, without degrading the pretrained models' core alignment or reasoning abilities.

Table 6: Efficiency analysis of our methods, which use the same prompt to calculate the first token generation time.

| Method | prefilling | decoding | total |
|---|---|---|---|
| Qwen2.5VL | 0.53 | 1.14 | 1.67 |
| Qwen2.5VL + Ours | 1.08 | 1.15 | 2.23 |
| Qwen2.5VL + CoT | 0.56 | 3.44 | 4.00 |

Table 7: Ablation about the effectiveness of ZoomText on Qwen2.5-VL-3B.

| Methods | TextHalu-Bench | $STVQA_{Test}$ | $TextVQA_{Val}$ | AI2D | $OCRVQA_{CORE}$ | $SEEDBench_{Text}$ | $GOT_{Scene}$ |
|---|---|---|---|---|---|---|---|
| Baseline | 48.3 | 67.3 | 79.1 | 78.1 | 70.2 | 66.7 | 85.2 |
| with text detector | 53.4 | 67.9 | 80.3 | 78.2 | 70.4 | 67.9 | 85.8 |
| w/o Glimpse | 52.9 | 67.3 | 78.8 | 78.2 | 69.8 | 70.2 | 85.2 |
| w/o Refocus | 53.5 | 67.3 | 78.9 | 78.2 | 69.8 | 70.2 | 85.1 |
| **Ours** | **53.8** | **67.6** | **80.3** | **78.2** | **70.5** | **70.2** | **86.0** |

**Efficiency Analysis.** As a training-free method, we report the inference time overhead introduced by our approach. As shown in Tab. 6, our method inevitably incurs additional computation in the prefilling stage, where attention maps from all layers are extracted and stored before decoding. However, we argue that this overhead is acceptable, as our approach remains more efficient than other test-time scaling methods such as Chain-of-Thought prompting (introduced in Sec. F). Furthermore, our method does not require any additional modules or external models to assist decoding, maintaining a streamlined and lightweight inference process.

# F More Ablation Studies

**Additional Implementation Details: Comparison with Other Hallucination Mitigation Methods.**

*(1) Adversarial Training.* To construct adversarial training data, we employ the image-text editing tool **TextCtrl** [93] to synthetically perturb the textual content of images from existing scene text datasets, including *CTW1500* [105], *ICDAR 2015* [25], and *TotalText* [106]. Following a targeted editing strategy, we generate up to three adversarial variants per image, depending on the number of text instances it contains, as shown in Fig. 9. Editing operations include character-level insertions, deletions, substitutions, and replacements with visually similar but semantically misleading characters. This process yields approximately **10,000** adversarial image-text pairs designed to introduce non-semantic visual perturbations that challenge both grounding and recognition. We fine-tune both *Mini-Monkey* and *Qwen2.5-VL* on the augmented dataset, using the original fine-tuning hyperparameters and training for one epoch. The fine-tuned models are then directly evaluated on downstream benchmarks to assess their robustness against semantic hallucination.

*(2) Chain-of-Thought (CoT) Prompting.* Our CoT-based hallucination mitigation strategy follows a two-stage inference process. In the first stage, the model generates an initial answer using standard inference procedures. In the second stage, we feed the model with both the original image and its previously generated answer, along with a CoT-style prompt that explicitly instructs the model to reflect on and verify its initial prediction by more carefully grounding it in the visual text regions. We design the following Chain-of-Thought (CoT) prompt to guide the second-stage reasoning:

> *"Your previous answer was: '{{answer}}'. Please carefully examine the text in the image again and verify whether the answer is fully supported by the visual evidence. If necessary, correct the answer based on the actual content in the image."*

Therefore, as further demonstrated in Tab. 8, Qwen2.5-VL also exhibits consistent improvements, providing additional evidence of the effectiveness of our approach.

**Additional Effectiveness Results of ZoomText.** As shown in Tab. 7, the experimental results further validate the effectiveness of ZoomText. In particular, incorporating both the Glimpse and ReFocus modules leads to notable performance gains, demonstrating the benefit of our progressive refinement

Table 8: Comparison of different hallucination mitigation methods on Qwen2.5-VL-3B.

| Methods | TextHalu-Bench | STVQA$_{Test}$ | TextVQA$_{Val}$ | GOT$_{Scene}$ | OCRVQA$_{CORE}$ | SEEDBench$_{Text}$ | AI2D |
|---------|---------------|----------------|-----------------|---------------|-----------------|-------------------|------|
| Baseline | 48.3 | 67.3 | 79.1 | 78.1 | 70.2 | 66.7 | 85.2 |
| Adv. | 49.1 | 67.2 | 78.6 | 78.1 | 70.4 | 67.9 | 85.2 |
| CoT | 48.5 | 67.7 | 79.4 | 78.2 | 70.4 | 70.2 | 85.5 |
| Ours | **53.8** | 67.6 | **80.3** | 78.2 | **70.5** | 70.2 | **86.0** |

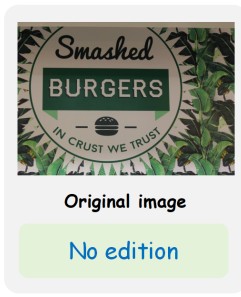 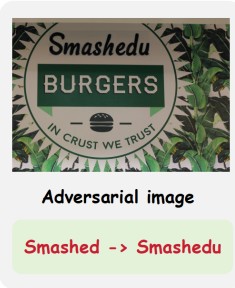 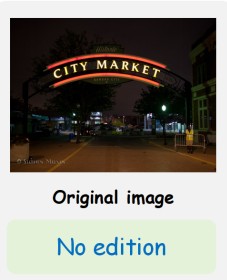 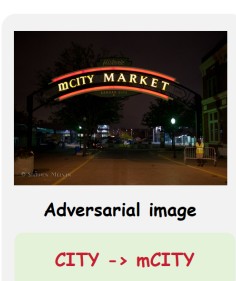

Original image — No edition

Adversarial image — Smashed -> Smashedu

Original image — No edition

Adversarial image — CITY -> mCITY

Figure 9: Visualization of adversarial training data.

strategy for region localization. By incrementally narrowing the model's attention to relevant visual areas, ZoomText effectively reduces ambiguity in visual-language alignment. These findings highlight the importance of hierarchical attention in improving scene text spotting and understanding.

Our proposed ZoomText module is based on two core assumptions: **(1) that query-to-image attention can effectively highlight relevant visual regions such as signs or posters**, and **(2) that background or non-semantic tokens exhibit relatively stable attention dynamics across layers.** To empirically validate these assumptions, we conducted the following analyses:

First, to assess the effectiveness of ZoomText in capturing relevant text regions, we manually annotated 100 samples across TextHalu-Bench, TextVQA, and ST-VQA, with quadrilateral bounding boxes marking the target text areas. We then performed an **IoU-based ablation study** on Qwen2.5-VL-3B, comparing three variants: **(a) a baseline selecting top-k tokens using final-layer attention**, **(b) baseline + Glimpse (which uses query-to-image attention, Eq. 4)**, and **(c) baseline + Glimpse + Refocus (which incorporates attention variation dynamics, Eq. 5)**. As shown in Tab. 9, results show a **consistent improvement in IoU scores**, confirming the benefit of each step in refining the focus on relevant visual regions.

Second, to validate the Refocus assumption, we computed the **coefficient of variation (CV)** of attention scores across layers for each visual token. Tokens were categorized into: **(i) foreground (within bounding boxes)**, and **(ii) background (outside boxes but with high attention).** We compute the CV for each token $i$, defined as:

$$\text{CV}_i = \frac{\text{Std}(\alpha_1^i, \alpha_2^i, \ldots, \alpha_L^i)}{\text{Mean}(\alpha_1^i, \alpha_2^i, \ldots, \alpha_L^i)} \tag{7}$$

where $\alpha_L^i$ is the attention score of token $i$ at layer $L$. The analysis reveals that **foreground tokens exhibit significantly higher attention variation across layers (higher CV)**, while **background tokens remain relatively stable (lower CV)**, supporting our hypothesis that "sink tokens" can be identified through their consistent attention profiles.

**Analysis of Weighting Strategies for Cross-Layer Hidden State Fusion.** To mitigate the limitations of relying solely on the final output layer for text recognition and understanding, we adopt a weighted fusion strategy that combines hidden states from different transformer layers, modulated by a fusion coefficient $\lambda$. We perform a grid search over $\lambda \in \{0.1, 0.2, 0.4, 0.6, 0.8\}$ to investigate its effect on performance. As shown in Tab. 10, the optimal performance is achieved at $\lambda = \mathbf{0.1}$, resulting in an average accuracy improvement of 1.67% over the baseline.

Moreover, the results reveal a nuanced trade-off: when the selected hidden layer carries richer visual information, higher values of $\lambda$ (e.g., $\lambda = 0.6$ or 0.8) tend to improve performance on text spotting tasks. However, these higher weights lead to diminished performance in text understanding tasks,

Table 9: Ablation studies of ZoomText (**Left**) and attention variation analysis (**Right**).

| Method | Mini-Monkey | Qwen2.5-VL |
|---|---|---|
| Baseline | 42.3 | 46.1 |
| + Glimpse | 45.9 | 48.9 |
| + Refocus | 47.8 | 52.7 |

| Token Type | Mini-Monkey | Qwen2.5-VL |
|---|---|---|
| Foreground | 2.76 | 2.86 |
| Background | 0.00054 | 0.00051 |

Table 10: Analysis of weights for cross-layer hidden state fusion on Qwen2.5VL-3B.

| Weights | TextHalu-Bench | $STVQA_{Test}$ | $TextVQA_{Val}$ | AI2D | $OCRVQA_{CORE}$ | $SEEDBench_{Text}$ | $GOT_{Scene}$ |
|---|---|---|---|---|---|---|---|
| 0 | 48.3 | 67.3 | 79.1 | 78.1 | 70.2 | 66.7 | 85.2 |
| 0.1 | **53.8** | **67.6** | **80.3** | 78.2 | **70.5** | 70.2 | 86.0 |
| 0.2 | 53.4 | 66.7 | 77.7 | 78.3 | 67.9 | 69 | 86.2 |
| 0.4 | 50.7 | 63.5 | 73.2 | 80 | 62 | 72.6 | **87.2** |
| 0.6 | 45 | 58.2 | 65.5 | **80.3** | 56.9 | **76.2** | 85.4 |
| 0.8 | 27.2 | 42.5 | 45.7 | 80.1 | 43 | 69 | 35.8 |

Table 11: Analysis of layers for hidden state fusion on Qwen2.5VL-3B.

| Layer Index | TextHalu-Bench | $STVQA_{Test}$ | $TextVQA_{Val}$ | AI2D | $OCRVQA_{CORE}$ | $SEEDBench_{Text}$ | $GOT_{Scene}$ |
|---|---|---|---|---|---|---|---|
| 0-10 | 52.5 | 67.3 | 78.6 | 78.2 | 69.7 | 69 | 86.1 |
| 10-20 | 52.9 | 67.4 | 78.9 | 78.2 | 70 | 70.2 | 85.4 |
| 20-35 | 53.1 | 67 | 79 | 78.1 | 70.4 | 69 | 85 |
| random | 53.1 | 67.1 | 78.9 | 78.2 | 69.9 | 70.2 | 85.4 |
| ours | 53.8 | 67.6 | 80.3 | 78.2 | 70.5 | 70.2 | 86.0 |

likely due to an overemphasis on visual features at the expense of semantic comprehension. This finding underscores the importance of carefully balancing visual and linguistic cues in cross-layer fusion for different types of downstream tasks.

**Analysis of Hidden Layer Contributions to Fusion.** To better understand which layers are most beneficial for visual grounding, we conduct an ablation study examining the effects of fusing hidden states from different model depths. Specifically, we compare five fusion strategies: (a) early layers (layers 0–10), (b) middle layers (layers 10–20), (c) late layers (layers 20–35), (d) randomly selected layers, and (e) our proposed layer selection method. As shown in Tab. 11, fusing hidden states from our selected layers yields the most substantial performance gain. These results indicate that not all layers contribute equally to grounding, and that a carefully chosen subset can more effectively capture the visual information necessary for hallucination mitigation.

