# OpenReview forum: "When Semantics Mislead Vision: Mitigating Large Multimodal Models Hallucinations in Scene Text Spotting and Understanding"
_NeurIPS.cc/2025/Conference — NeurIPS 2025 poster_

### Official Review · Reviewer_wv1z · 2025-06-04

**Clarity:** 3
**Significance:** 3
**Originality:** 3
**Rating:** 4
**Confidence:** 4

**Summary:**

This paper investigates the phenomenon of semantic hallucination in Large Multimodal Models (LMMs), specifically within scene text spotting and understanding tasks. The authors identify that LMMs often rely on semantic priors and fail to recognize visually ambiguous or non-semantic text accurately. To address this, the paper proposes a training-free framework with two key components: (1) ZoomText, a coarse-to-fine text region estimator, and (2) Grounded Layer Correction (GLC), which guides decoding by leveraging transformer layers with better visual grounding. A new benchmark, TextHalu-Bench, is also introduced to evaluate hallucination behavior in LMMs. Experiments across several benchmarks demonstrate the effectiveness and generalizability of the proposed approach.

**Questions:**

**1.Incomplete Evaluation on OCR Benchmarks:**
The proposed method is not evaluated on MME or MME-VideoOCR, both of which include OCR-relevant tasks and are widely used for assessing multimodal LLMs. Given that the method aims to mitigate semantic hallucinations involving visual text, such evaluation is necessary to verify its robustness and generalizability across diverse scenarios.

**2.Unverified Methodological Assumptions:**
The method assumes that visually prominent text (e.g., signs or posters) draws model attention, and that background tokens have stable attention patterns across layers. However, no analysis or visualization is provided to support these assumptions. Without empirical validation, the method’s motivation remains speculative and its design less convincing.

**Ethical Concerns:**

["NO or VERY MINOR ethics concerns only"]

**Final Justification:**

This is a well-structured and thoughtful rebuttal. The authors have effectively addressed the weaknesses I raised.

Reviewer wV1z
I appreciate the authors' thorough response to my feedback. The rebuttal effectively addresses the two main weaknesses I raised.

Regarding Weakness 1, the authors have provided new experimental results on both the MME and MME-VideoOCR benchmarks. These results, which demonstrate consistent performance improvements, significantly strengthen the claim that the proposed method is robust and generalizable. I am satisfied with this additional evaluation.

Regarding Weakness 2, the authors have provided empirical validation for their key methodological assumptions. The new ablation study, which uses IoU scores to evaluate the effectiveness of the ZoomText module, and the analysis of the attention coefficient of variation (CV) for foreground versus background tokens directly address my concerns. These results provide clear evidence supporting the design choices of the proposed method and successfully move the motivation from speculative to empirically grounded.

Overall, the authors' response has successfully resolved the weaknesses I identified. My initial assessment remains unchanged, and I maintain my recommendation of borderline accept.

**Limitations:**

Yes

**Quality:**

3

**Strengths And Weaknesses:**

**Strength:**

**1.Novelty:**
The work tackles the underexplored issue of semantic hallucination in OCR scenarios, distinct from prior object or attribute hallucination studies. The concept of layer-wise hallucination tendency analysis and visual grounding correlation is original and empirically well-motivated.

**2.Motivation and Clarity:**
The paper provides a clear and compelling motivation by identifying a failure mode in existing LMMs. The problem is illustrated with intuitive examples and well-analyzed causes rooted in LLM architecture.

**3.Effectiveness:**
Across multiple benchmarks, the method consistently improves performance, especially on hallucination-prone samples. Gains are shown to be more pronounced on models with stronger OCR capabilities, which aligns with the method’s design.

**Weakness:**

**1. Incomplete Evaluation on OCR Benchmarks:**
The MME benchmark,a widely used general-purpose benchmark for multimodal large models,includes OCR-related tasks. Therefore, the proposed method should also be evaluated on MME and MME-VideoOCR to more rigorously assess its effectiveness in mitigating semantic hallucinations. Without such evaluation, the generalizability and robustness of the method across broader multimodal scenarios remain uncertain.

[1] MME: A Comprehensive Evaluation Benchmark for Multimodal Large Language Models. CoRR 2023

[2] MME-VideoOCR: Evaluating OCR-Based Capabilities of Multimodal LLMs in Video Scenarios. CoRR 2025

**2. Unverified Methodological Assumptions:**
The design of ZoomText relies on two key assumptions: (1) that signs, posters, or product packaging naturally attract model attention during question answering; and (2) that background or non-semantic tokens exhibit relatively stable attention patterns across layers. However, the paper does not present any experiments or visualizations to directly validate these assumptions. Without empirical support, the motivation behind the proposed method remains speculative. This lack of validation undermines the credibility of the approach and weakens the explanatory power of its design.

---

> ### Author Rebuttal · Authors · 2025-07-31
>
> # To reviewer wv1z
>
> We thank the reviewer for the detailed and encouraging feedback. We appreciate your recognition of the novelty of our work in addressing the underexplored issue of semantic hallucination in OCR scenarios, which is distinct from previous studies on object or attribute hallucination. We're glad you found the analysis of layer-wise hallucination tendency and its correlation with visual grounding both original and empirically sound. We also value your positive comments on the clarity and motivation of the paper, particularly in highlighting a failure mode in existing LMMs through intuitive examples and architectural insights. In addition, we’re encouraged by your acknowledgment of the method’s consistent performance improvements across benchmarks, especially on hallucination-prone samples, and how these gains align well with the method’s design.
>
> Below, we respond to each Weakness (W) or Question (Q) with a corresponding Answer (A):
>
> ## [W1/Q1]: Incomplete Evaluation on OCR Benchmarks
>
> [A1]: Thank you for your valuable feedback. We agree that incorporating more OCR-related benchmarks can better demonstrate the robustness and generalizability of our method. To address this, we have complemented our evaluation with experiments on both **MME and MME-VideoOCR** (a concurrent work). The results show:
>
> On MME, our method brings improvements in **perception-based subtasks such as OCR, counting, and positioning**. However, due to the binary (yes/no) nature of many MME questions, where models can often guess the answer, the absolute improvement is relatively modest.
>
> On MME-VideoOCR, we observe significant gains, especially in **complex scene-text-centric video question answering tasks**, where visual text grounding is more challenging.
>
> These results, along with the existing experiments (e.g., Table 5), provide strong evidence of the robustness and general applicability of our proposed method across diverse OCR and multimodal reasoning benchmarks. We will include these new results in the revised version of the paper.
>
> | Task                   | minimonkey | minimonkey+ours | qwen2.5vl | qwen2.5vl+ours |
> |------------------------|------------|------------------|-----------|----------------|
> | perception             | 1451.78    | 1461.91          | 1567.70   | 1572.2        |
> | reasoning              | 427.5      | 427.5            | 601.07   | 601.07          |
> | OCR                    | 95         | 98               | 132.5     | 136             |
> | artwork                | 146.25     | 146.25           | 143.75    | 143.75         |
> | celebrity              | 118.529    | 118.529          | 127.35   | 127.35        |
> | color                  | 168.333    | 168.333          | 170.0   | 170.0        |
> | count                  | 143.333    | 148.0          | 155.0   | 156.0        |
> | existence              | 190        | 190              | 185.0    | 185.0            |
> | landmark               | 156.5      | 156.5            | 177.0     | 177.0          |
> | position               | 153.333    | 155.8          | 151.67   | 151.67        |
> | posters                | 129.252    | 129.252          | 171.43   | 171.43        |
> | scene                  | 151.25     | 151.25           | 154.0    | 154.0         |
> | code_reasoning         | 92.5       | 92.5             | 155.0    | 155.0           |
> | commonsense_reasoning  | 115        | 115              | 128.57 | 128.57            |
> | numerical_calculation  | 42.5       | 42.5             | 125.0    | 125.0           |
> | text_translation       | 177.5      | 177.5            | 192.5     | 192.5          |
> (Experiment on MME)
>
>
>
> | Task      | Qwen2.5-VL-7B | Qwen2.5-VL-7B+ours |
> |-----------|-----------|----------------|
> | TR        | 70.1      | 70.1           |
> | VTQA      | 70.2      | 73.4           |
> | TG        | 57.6      | 58.9           |
> | AR        | 69.2      | 69.2           |
> | CDT       | 46.8      | 46.8           |
> | STP       | 66.4      | 66.4           |
> | CFTU      | 17.1      | 17.3           |
> | TBR       | 49.8      | 49.6           |
> | TBVU      | 52.5      | 52.5           |
> | RVT       | 71.3      | 71.3           |
> | **Total** | **59.2**  | **60.8**       |
> (Experiment on MME-VideoOCR)
>
>
> ## [W2/Q2]: Unverified Methodological Assumptions.
> [A2]: Thank you for pointing out this important aspect. Our proposed ZoomText module is based on two core assumptions: **(1) that query-to-image attention can effectively highlight relevant visual regions such as signs or posters**, and **(2) that background or non-semantic tokens exhibit relatively stable attention dynamics across layers.** To empirically validate these assumptions, we conducted the following analyses:
>
>
> First, to assess the effectiveness of ZoomText in capturing relevant text regions, we manually annotated 100 samples across TextHalu-Bench, TextVQA, and ST-VQA, with quadrilateral bounding boxes marking the target text areas. We then performed an **IoU-based ablation study** on Qwen2.5-VL-3B, comparing three variants: **(a) a baseline selecting top-k tokens using final-layer attention**, **(b) baseline + Glimpse (which uses query-to-image attention, Eq. 4)**, and **(c) baseline + Glimpse + Refocus (which incorporates attention variation dynamics, Eq. 5).** Results show a **consistent improvement in IoU scores**, confirming the benefit of each step in refining the focus on relevant visual regions.
>
>
>
> Second, to validate the Refocus assumption, we computed the **coefficient of variation (CV)** of attention scores across layers for each visual token. Tokens were categorized into: **(i) foreground (within bounding boxes)**, and **(ii) background (outside boxes but with high attention).** We compute the CV for each token $i$, defined as:
>
> $
> \mathrm{CV}_i = \frac{\mathrm{Std}(\alpha^i_1, \alpha^i_2, \dots, \alpha^i_L)}{\mathrm{Mean}(\alpha^i_1, \alpha^i_2, \dots, \alpha^i_L)}
> $
>
> where $\\alpha^i\_L$  is the attention score of token $i$ at layer $L$. The analysis reveals that \*\*foreground tokens exhibit significantly higher attention variation across layers (higher CV)\*\*, while \*\*background tokens remain relatively stable (lower CV)\*\*, supporting our hypothesis that "sink tokens" can be identified through their consistent attention profiles.
>
>
> Due to rebuttal constraints, we are unable to include visualizations at this stage, but we will incorporate these empirical results and more qualitative examples into the revised version of the paper to strengthen the methodological soundness of our design.
>
>
>
> | Method      | Mini-Monkey | Qwen2.5-VL |
> |-------------|-------|-------|
> | Baseline    | 42.3  | 46.1  |
> | + Glimpse   | 45.9  | 48.9  |
> | + Refocus     | 47.8      | 52.7  |
> (ablation studies of ZoomText )
>
>
> | Method      | Mini-Monkey | Qwen2.5-VL |
> |-------------|-------|-------|
> | foreground tokens    | 2.76  | 2.86  |
> | background tokens   | 0.00054  | 0.00051  |
> (abalation studies of relative attention)

---

> > ### Comment · Reviewer_wv1z · 2025-08-01
> >
> > This is a well-structured and thoughtful rebuttal. The authors have effectively addressed the weaknesses I raised.
> >
> > Reviewer wV1z
> > I appreciate the authors' thorough response to my feedback. The rebuttal effectively addresses the two main weaknesses I raised.
> >
> > Regarding Weakness 1, the authors have provided new experimental results on both the MME and MME-VideoOCR benchmarks. These results, which demonstrate consistent performance improvements, significantly strengthen the claim that the proposed method is robust and generalizable. I am satisfied with this additional evaluation.
> >
> > Regarding Weakness 2, the authors have provided empirical validation for their key methodological assumptions. The new ablation study, which uses IoU scores to evaluate the effectiveness of the ZoomText module, and the analysis of the attention coefficient of variation (CV) for foreground versus background tokens directly address my concerns. These results provide clear evidence supporting the design choices of the proposed method and successfully move the motivation from speculative to empirically grounded.
> >
> > Overall, the authors' response has successfully resolved the weaknesses I identified. My initial assessment remains unchanged, and I maintain my recommendation of borderline accept.

---

> > > ### Author Response · Authors · 2025-08-06
> > > **Response to the comment**
> > >
> > > Thank you very much for your positive feedback and thoughtful follow-up. We truly appreciate your recognition of our efforts to address the concerns and your continued support of our work.

---

> > > ### Author Response · Authors · 2025-08-06
> > > **Response to the comment**
> > >
> > > Thank you very much for your positive feedback and thoughtful follow-up. We truly appreciate your recognition of our efforts to address the concerns and your continued support of our work.

---

### Official Review · Reviewer_6wHJ · 2025-06-20

**Clarity:** 2
**Significance:** 3
**Originality:** 3
**Rating:** 4
**Confidence:** 4

**Summary:**

This paper identifies and addresses a specific failure mode in Large Multimodal Models (LMMs) which the authors term “semantic hallucination.” This phenomenon occurs when LMMs, confronted with visually ambiguous or non-semantic scene text, generate semantically plausible but visually incorrect answers by relying on language priors instead of visual evidence. The authors first conduct an analysis and find a strong negative correlation between a Transformer layer’s tendency to hallucinate and the amount of attention it allocates to the ground-truth text regions.

Based on this insight, they propose a novel, training-free framework to mitigate this issue. The framework consists of two main components: (1) ZoomText, a detector-free, coarse-to-fine strategy to localize potential text regions within an image, and (2) Grounded Layer Correction (GLC), a mechanism that adaptively selects the most visually-grounded layer and fuses its representations into the final decoding step to correct potential hallucinations.

**Questions:**

- Regarding the claim of a potential causal link between attention and hallucination, could the authors elaborate on this? Is it possible that a layer’s strong attention to a text region is a consequence of it correctly identifying the tokens, rather than the cause of the correct identification? How might one design an experiment to further probe this causal relationship?
- The fusion weight w in GLC is set to 0.1, and the top-K in ZoomText is 128. How sensitive is the performance to the choice of K? Does the optimal K vary significantly across different models or image resolutions? A brief discussion on the robustness or a principled way to set these parameters would be beneficial.
- The method shows impressive gains, but what are the primary remaining failure modes? Does the framework sometimes “over-correct” and harm the recognition of highly semantic or idiomatic text? Or does it primarily fail on extremely distorted or occluded non-semantic text where even the best internal layer struggles? Providing a few qualitative examples of failure cases in the appendix would offer a more complete picture of the method’s boundaries.
- The efficiency analysis in Table 6 transparently shows the inference overhead during the prefilling stage. Could you contextualize this for a real-world application? For instance, how does this overhead compare to the overall latency of generating a long-form answer? Is the prefilling stage a one-time cost per query, regardless of the generated answer length? Clarifying this could help readers assess the practical trade-offs.

**Ethical Concerns:**

["NO or VERY MINOR ethics concerns only"]

**Final Justification:**

The authors have solved my concerns. I believe this is a weak-accept-level paper. So I suggest that AC and PC decide on this paper based on the batch situation.

**Limitations:**

yes

**Quality:**

3

**Strengths And Weaknesses:**

### **Strengths**
- The paper tackles a timely and significant problem. The concept of “semantic hallucination” in the context of scene text is a novel and precise framing of a critical weakness in current LMMs.
- The proposed solution is both clever and well-motivated by the initial analysis. The core finding that links hallucination tendency to layer-wise attention provides a solid, empirical foundation for the entire framework.
- The creation of TextHalu-Bench is a significant contribution in itself. It provides the community with a much-needed tool to measure a specific, challenging aspect of LMM performance that is overlooked by existing benchmarks. Figure 4 clearly justifies its necessity.

### **Weaknesses**
- The paper states that its analysis suggests “a causal relationship between accurate attention allocation and the mitigation of semantic hallucination.” While the negative correlation is strong and provides a compelling motivation, **claiming causality might be an overstatement**. It is also plausible that both poor attention allocation and hallucination are symptoms of a common underlying cause (e.g., the model’s failure to recognize the region as containing text from the outset). A more tempered discussion of this correlation would strengthen the scientific claim.
- The results on LLaVA-NeXT show a marginal improvement, which the authors rightly attribute to its limited base OCR capabilities. This highlights a key limitation: the method is a correction mechanism, not a fundamental capability enhancer. **It can help a model that can see the text to report it faithfully, but it cannot help a model that is effectively blind to the text**. This dependency could be discussed more prominently as a scope-defining weakness of the approach.
- **Heuristic Nature of Refocus**: The “Refocus” step is based on the heuristic that background or “register” tokens exhibit stable attention patterns across layers. While the ablation shows this works well, the paper could benefit from a deeper analysis of this phenomenon. Visualizing which tokens are actually filtered out by this step would provide stronger evidence for the hypothesis.

---

> ### Author Rebuttal · Authors · 2025-07-31
>
> # To reviewer 6wHJ
>
> Thank you for the thoughtful feedback. We're glad you found our focus timely and novel, and appreciated both the framing and attention analysis. We also appreciate your recognition of TextHalu-Bench.
>
> Below, we address each Weakness (W) or Question (Q) with a corresponding Answer (A):
>
> ## [W1/Q1]: Discussion about the causal relatioinship between visual attention grounding and halluction generation.
>
> **[A1]**: Thank you for the thoughtful question. We agree that claiming a direct **causal relationship** may be too strong, and we appreciate your suggestion to phrase it more cautiously. It is indeed plausible that both poor attention and hallucination stem from a shared issue—such as the model’s failure to recognize meaningful text.
>
> That said, we believe it's valuable to explore whether **inaccurate attention contributes to hallucination**, given the **auto-regressive decoding** process in LLMs. Misalignment during the **encoding stage** could bottleneck downstream reasoning.
>
> To examine this, we ran a new ablation on 100 samples from TextHalu-Bench, TextVQA, and ST-VQA. We manually annotated ground-truth text boxes and **artificially reallocated attention** from these to background regions using:
>
> $$
> \\alpha' = \\frac{(1 - \\lambda) \\cdot \\alpha_{\\text{gt}} + \\lambda \\cdot \\alpha_{\\text{bg}}}{\\sum \\left[(1 - \\lambda) \\cdot \\alpha_{\\text{gt}} + \\lambda \\cdot \\alpha_{\\text{bg}}\\right]}
> $$
>
> Here, $\alpha_{\text{gt}}$ and $\alpha_{\text{bg}}$ denote the original attention scores over ground-truth and background tokens, respectively.  The denominator computes the total adjusted attention score to ensure that $\alpha'$ remains a valid probability distribution.
> As **λ** increases, hallucination scores rise significantly (same metric as Fig. 2), confirming that **misallocated attention worsens hallucinations**.
>
> We will incorporate this analysis in the revised paper and revise our claim from a **“causal relationship”** to a **“strong correlation”**, ensuring a more accurate and robust interpretation.
>
> | λ (Attention Shift Ratio) | Hallucination Score |
> |---------------------------|---------------------|
> | 0.1                       | 0.36                |
> | 0.3                       | 0.38                |
> | 0.5                       | 0.54                |
> | 0.7                       | 0.62                |
>
> ## [W2]: Discussion about the model`s ability limitation.
>
> [A2]: Thank you for raising this important point. We agree that our method functions as a correction mechanism, not a capability enhancer. Hallucinations in multimodal models broadly fall into two types:
> (1) Faithfulness Errors – the model perceives the information but responds inaccurately;
> (2) Knowledge Gaps – the model fails to perceive or understand the required input.
>
> Our training-free framework targets Type 1 errors and cannot improve models lacking fundamental OCR ability (e.g., LLaVA-NeXT). Still, we believe addressing faithfulness is a necessary step before tackling deeper knowledge gaps.
>
> We have noted this scope in Appendix "Limitations" (Lines 597–602) and will make it more explicit in the revised version. We also plan to explore external knowledge integration in future work to expand the framework’s coverage.
>
> ## [W3]: Heuristic Nature of Refocus.
>
> [A3]: We agree that a deeper analysis of the Refocus step would enhance the clarity and credibility of our method. Although we are unable to include visualizations during the rebuttal phase, we conducted two ablation studies to empirically support the underlying heuristics: 1. **IoU Evaluation**: On the same validation set in [A1]. We compared three configurations using Qwen2.5-VL: (1) Baseline: Selects top-$k$ tokens using attention from the last LLM layer. (2) +Glimpse: Utilizes query-to-image attention (Eq. 4) to select tokens. (3) +Refocus: Filters irrelevant tokens using relative attention dynamics (Eq. 5).
>
> The average IoU between predicted and ground-truth regions increases notably after adding Refocus, indicating improved localization precision.
>
> | Method      | Mini-Monkey | Qwen2.5-VL |
> |-------------|-------|-------|
> | Baseline    | 42.3  | 46.1  |
> | + Glimpse   | 45.9  | 48.9  |
> | + Refocus     | 47.8      | 52.7  |
>
> 2. **Coefficient of Variation (CV) Analysis**. To test the assumption that background tokens exhibit stable attention, we computed the coefficient of variation (CV) of attention scores across layers for each visual token:
>
> $
> \mathrm{CV}_i = \frac{\mathrm{Std}(\alpha^i_1, \alpha^i_2, \dots, \alpha^i_L)}{\mathrm{Mean}(\alpha^i_1, \alpha^i_2, \dots, \alpha^i_L)}
> $
>
> where $\\alpha^i\_L$ is the attention score of token $i$ at layer $L$. We categorized tokens into: (1) Foreground: Tokens inside bounding boxes; (2) Background: Tokens outside bounding boxes but with high attention
>
> Our analysis shows that foreground tokens have significantly higher CV, while background tokens remain stable (lower CV). This validates our heuristic: **stable tokens likely represent less semantically relevant areas and can be effectively filtered via Refocus.**
>
> | Method      | Mini-Monkey | Qwen2.5-VL |
> |-------------|-------|-------|
> | foreground tokens    | 2.76  | 2.86  |
> | background tokens   | 0.00054  | 0.00051  |
>
> ## [Q2]: The analysis of hyperparamter K.
> **[A2]**: We agree that understanding the role of **hyperparameter K** in ZoomText is essential for clarity and reproducibility. We conducted ablations on **Qwen2.5-VL** and **Mini-Monkey** across four datasets with varying image resolutions, leading to three key observations: 1. **Model-Agnostic Behavior**:   The optimal **K** generalizes well across architectures. Performance is not highly sensitive to the backbone. 2. **Robust Value Range**:
> While there's no fixed best value, performance remains stable when **K ∈ [96, 160]**. Small **K** may miss text; large **K** may introduce noise. We use **K = 128** as a balanced default. 3. **Resolution Robustness**:   Most datasets (≤1000px) show no performance drop with **K = 128**. On high-res images from **ReCTS** (>2000px), the optimal **K** shifts to **[160, 224]**, suggesting larger values help in ultra high-resolution scenarios. We acknowledge that **K** is empirically chosen. Still, its consistent performance across diverse datasets demonstrates robustness. We will highlight the lack of a resolution-aware selection method in the **Limitations** section.
>
> | K   |  TextHalu-Bench | TextVQA |  SEEDBench$_{Text}$  | OCRVQA
> |-----|---------|---------|-------|--------|
> | 64  | 50.1    | 78.8    | 67.9  | 70.0   |
> | 96  | 53.4    | 79.9    | 69.0  | 70.2   |
> | 128 |**53.8** |**80.3** |**70.2**|**70.5**|
> | 160 | 53.6    | 80.1    | 70.2  | 70.0   |
> | 192 | 52.1    | 80.1    | 70.2  | 70.0   |
> | 224 | 52.2    | 79.2    | 69.0  | 69.9   |
> （Experiments on Qwen2.5-VL）
>
> | K   |  TextHalu-Bench | TextVQA |  SEEDBench$_{Text}$  | OCRVQA
> |-----|---------|---------|-------|--------|
> | 64  | 48.9    | 74.8    | 83.3  | 38.9   |
> | 96  | 50.1    | 75.0    | 83.3  | 39.8   |
> | 128 |**50.6** |**75.0** |**84.5**|**39.9**|
> | 160 | 50.5    | 74.8    | 84.5  | 39.9   |
> | 192 | 49.9    | 74.1    | 83.3  | 38.2   |
> | 224 | 49.9    | 73.8    | 83.3  | 38.2   |
> （Experiments on MiniMonkey）
>
> | K   | Qwen2.5-VL
> |-----|------------|
> | 64  | 51.2       |
> | 96  | 51.6       |
> | 128 | 52.7       |
> | 160 | 53.1       |
> | 192 | **53.8**   |
> | 224 | 53.7       |
> (Experiments on High-resolution ReCTS-VQA)
>
> [1] Icdar 2019 robust reading challenge on reading chinese text on signboard. ICDAR 2019.
>
> ## [Q3]: Discussions about the failure cases.
>
> [A3]: Thank you for raising this important point. Due to the limitations of the rebuttal stage, we are unable to include visual failure case examples at this time, but we will provide more qualitative samples in the appendix of the revised version. Regarding failure modes, we clarify the following: (1) **Minimal Over-correction**: Our method does not impair recognition of semantic or idiomatic text. As shown in Table 1, OCR performance is slightly improved, indicating no adverse effects. (2) **Preserved Generalization**: Table 5 shows stable performance on general benchmarks, suggesting hallucination mitigation does not come at the cost of reasoning ability. (3) **Typical Failures Reflect Model Limits**: Errors mainly occur in inherently difficult cases—e.g., distorted/occluded text, rare scripts, or questions requiring external knowledge—where even base models struggle. As a training-free method, ours ensures faithful outputs but cannot recover missing knowledge.
>
> We will include detailed failure cases in the revised version to clarify the scope and limitations.
>
> ## [Q4]: Further explanation about efficiency.
>
> [A4]: Thank you for the insightful question. The added latency from our method occurs only during the prefilling stage, where intermediate attention scores are computed. This is a **one-time cost per query**, independent of the output length. In long-form generation tasks (e.g., captioning), our experiments with Qwen2.5-VL show that prefilling adds less than 35% to the total inference time when outputs exceed 50 tokens.
>
> This overhead is modest compared to the **dominant cost of autoregressive decoding**, especially for longer outputs. We thus consider the trade-off acceptable for real-world use.
>
> | Output token Length | Method    | Prefilling (s) | Decoding (s) | Total Time (s) |
> |---------------------|-----------|------------|----------|------------|
> | (0,20]              | Baseline  | 0.62       | 1.03     | 1.65       |
> |                     | Ours      | 1.22       | 1.04     | 2.26       |
> | (20,50]             | Baseline  | 0.65       | 1.92     | 2.57      |
> |                     | Ours      | 1.32       | 1.94     | 3.26      |
> | (50,]               | Baseline  | 0.66       | 2.64     | 3.30      |
> |                     | Ours      | 1.34       | 2.65     | 3.99      |
> （ablation studies on efficiency in 80G A-100 ）

---

> ### Author Response · Authors · 2025-08-06
> **Seek Further Response**
>
> Dear Reviewer 6wHJ,
>
> Thank you for your support and helpful comments. We've tried our best to address your concerns, and we hope our responses make sense to you. Importantly, we much value your comments and would be happy to discuss more. If you have any additional questions or open discussions, please don't be hesitant to leave more comments. We are always available at all time, to actively address any concerns or be prepared for more discussions.
>
> Your opinions are rather important for us to improve the work!
>
> Thank you!
>
> Sincerely,
>
> Authors

---

> > ### Comment · Reviewer_6wHJ · 2025-08-08
> >
> > You have solved most of my concerns. I choose to maintain my rating.

---

### Official Review · Reviewer_1eg9 · 2025-07-02

**Clarity:** 3
**Significance:** 3
**Originality:** 3
**Rating:** 4
**Confidence:** 4

**Summary:**

This paper tackles a problem in Large Multimodal Models (LMMs) called "semantic hallucination." This is when a model reads text based on what makes sense semantically, rather than what is actually in the image. For instance, it might see "MMOTEL" but incorrectly output "MOTEL" because that's a real word. The authors discovered a key insight: within the LMM, some internal layers are much better at focusing on the actual text regions and are therefore less likely to hallucinate. Based on this, they propose a training-free framework to fix the problem: First, it finds the text regions in the image using a coarse-to-fine approach, without needing an external tool. Then, Grounded Layer Correction identifies the "best-grounded" internal layer (the one focusing most on the text) and uses its information to correct the model's final output. To test their method, they also created a new benchmark called TextHalu-Bench. Experiments show their approach effectively reduces these hallucinations and improves the model's accuracy on scene text understanding tasks.

**Questions:**

* How is the arg max operation performed in line 190: l* = arg max_l A_l. Is it that the attention map for a layer, A_l, is a matrix? It would be great to provide more details about this.

**Ethical Concerns:**

["NO or VERY MINOR ethics concerns only"]

**Final Justification:**

Thanks to the authors for the response. It mostly addressed my concern and I will raise my score.

**Limitations:**

yes

**Quality:**

2

**Strengths And Weaknesses:**

Strength
* This work investigates the significant issue of 'semantic hallucination,' a phenomenon where Vision-Language Models exhibit a propensity to rely on their internal semantic priors to generate textual output, rather than faithfully grounding their responses in the visual evidence. The formalization and subsequent mitigation of this problem represent a valuable contribution toward enhancing the reliability of LMMs for real-world, text-centric applications.
* The creation of the TextHalu-Bench benchmark will be a valuable resource for future research in this area.

Weakness
* The methodology described for Equation (2) seems to contradict the fundamental principles of auto-regressive models. In line 132, The paper states "feed both the hallucinated token y_hal and the ground-truth token y_gt into the LMM's output head." This means that it feeds both y_hal and y_gt into the output head to produce their respective probabilities. How is this reconciled with the fact that an auto-regressive model's hidden state at a given step is determined only by its previous inputs, not by the token it is about to predict?
* While OCRBench is used for analysis in Figure 1 and mentioned in Section 3.2, its results are conspicuously absent from the main experimental results in Table 1 (Section 5.2). Could the authors clarify why this benchmark was excluded from the final evaluation and it would be helpful to include it in the evaluation.

---

> ### Author Rebuttal · Authors · 2025-07-31
>
> # To reviewer 1eg9
>
> We thank the reviewer for the detailed and encouraging feedback. We're glad that you recognized the significance of the semantic hallucination issue, particularly the tendency of Vision-Language Models to rely on internal semantic priors rather than grounding their outputs in visual evidence. We appreciate your acknowledgment of our formalization and mitigation of this problem as an important step toward improving the reliability of LMMs in real-world, text-centric applications. We're also pleased that you found the introduction of the TextHalu-Bench benchmark to be a valuable resource for future research in this area.
>
> Below, we respond to each Weakness (W) or Question (Q) with a corresponding Answer (A):
>
>
>
> ## [W1]: The concern of the hallucination score calculation.
>
> [A1]:  Thank you for your thoughtful comment. We agree that the wording in Line 132 may have caused some confusion. Specifically, the phrase “feed both the hallucinated token $y\_{\\text{hal}}$ and the ground-truth token $y\_{\\text{gt}}$ into the LMM's output head” was not precise.
>
> What we intended to convey is that, at each decoding step $t$, the model computes a probability distribution over the entire vocabulary based on the prefix $x\_{\\leq t}$. From this distribution, we **extract (not feed) the probabilities assigned to both $y\_{\\text{hal}}$ and $y\_{\\text{gt}}$,** as they are among the candidate tokens.
>
>
> This process is fully consistent with the autoregressive property of LLMs: the model’s hidden state is determined solely by the preceding tokens, and the comparison between $y\_{\\text{hal}}$ and $y\_{\\text{gt}}$ is made at the output level, not by modifying the input sequence or hidden states.
>
>
> We will revise the original sentence to clarify this distinction and appreciate the opportunity to improve the presentation.
>
>
>
>
> ## [W2]:  The absence of OCRBench in the main experiment.
>
> [A2]: Thank you for the suggestion. OCRBench was not included in the main results table because it primarily **evaluates text recognition from cropped text regions**, where scene text understanding plays a limited role. In contrast, our method is designed for open-world scene text spotting and understanding.
>
>
>
> Nonetheless, to provide a more comprehensive evaluation, we have conducted additional experiments on OCRBench. The results show **consistent improvements across multiple LMMs**, especially in scene-text-related VQA subtasks, confirming the generalization ability of our method. We will include these results in the revised version of the paper.
>
>
>
>
> | Model                 | Text Recognition | Scene Text-centric VQA | Doc-oriented VQA | Key Info Extraction | Handwritten Math Exp. | Final Score |
> |-----------------------|------------------|--------------------------|-------------------|----------------------|------------------------|--------------|
> | LLaVA-NeXT            | 191              | 156                      | 84                | 100                  | 0                      | 531          |
> | LLaVA-NeXT + Ours     | 188              | 159                      | 85                | 105                  | 0                      | 537          |
> | Qwen2.5-VL            | 249              | 172                      | 162               | 180                  | 60                     | 823          |
> | Qwen2.5-VL + Ours     | 250              | 177                      | 164               | 181                  | 60                     | 832          |
> | Mini-Monkey           | 250              | 168                      | 141               | 168                  | 71                     | 798          |
> | Mini-Monkey + Ours    | 250              | 175                      | 139               | 169                  | 71                     | 804          |
>
>
> ## [Q1]:  The confusion about how to select the best layer.
>
> [A1]: Thanks for your question. As defined in Equation (3), $A\_\\ell$ is a scalar attention score that quantifies how much the $\\ell$-th layer attends to the scene text regions.
>
> We compute $A\_\\ell$ for each layer and collect them into a 1D array $\\{A\_1, A\_2, \\dots, A\_L\\}$. We then apply the $\\arg\\max$ operation:
>
>
> $$
> \ell^\star = \arg\max_\ell A_\ell
> $$
>
>
> to select the layer that focuses most strongly on the text regions.

---

> > ### Comment · Reviewer_1eg9 · 2025-08-06
> >
> > Thanks to the authors for the response. It mostly addressed my concern and I will raise my score.

---

> > > ### Author Response · Authors · 2025-08-07
> > > **Official Comment by Authors**
> > >
> > > We sincerely appreciate your time and effort throughout the review. Thank you again for your valuable comments to improve the quality of our manuscript!

---

> ### Author Response · Authors · 2025-08-06
> **Seek Further Response**
>
> Dear Reviewer 1eg9,
>
> Thank you for your support and helpful comments. We've tried our best to address your concerns, and we hope our responses make sense to you. Importantly, we much value your comments and would be happy to discuss more. If you have any additional questions or open discussions, please don't be hesitant to leave more comments. We are always available at all time, to actively address any concerns or be prepared for more discussions.
>
> Your opinions are rather important for us to improve the work!
>
> Thank you!
>
> Sincerely,
>
> Authors

---

### Official Review · Reviewer_LbB3 · 2025-07-03

**Clarity:** 3
**Significance:** 3
**Originality:** 3
**Rating:** 4
**Confidence:** 3

**Summary:**

This paper presents “semantic hallucination”, a hallucination problem in MLLMs where the target task answer that appears in the input image is not semantically relevant (e.g., OCR on a text that has a typo), causing MLLMs to hallucinate a semantically relevant answer based on its strong language prior in their pre-trained LLMs. The author(s) presented an investigation of the underlying causes of this issue through cross-attention visualization and layer-wise token outputs likelihoods, and presents a mitigation framework that automatically augments hidden states within the LLM to reduce semantic hallucination. In addition, the author(s) also introduce TextHalu-Bench, a new benchmark targeting this particular issue, and show that the proposed framework is both effective in reducing hallucination and improving scene-text understanding tasks.

**Questions:**

Based on the above weaknesses, I have the following questions:

- Can the author(s) report the proposed method’s performance on knowledge-based, non-scene-text benchmarks, and compare them against baselines, such as OK-VQA [1]?
- Can the author(s) investigate the effectiveness of ZoomText in extracting areas of interest in the image?
- How frequently does Grounded Layer Correction change the proposed method?

I am currently on the fence about this paper. However, if the author(s) can conduct a more thorough exploration of the capabilities and limitations of the proposed method, I will be willing to raise my score.

Reference:
[1] https://okvqa.allenai.org/

**Ethical Concerns:**

["NO or VERY MINOR ethics concerns only"]

**Final Justification:**

All my concerns were addressed by the author(s) in the rebuttal, and thus my score was raised.

**Limitations:**

yes

**Quality:**

2

**Strengths And Weaknesses:**

**Strengths**:

**Presentation of an important hallucination issue and contribution of the associated benchmark**: As MLLMs are increasingly used to conduct text-rich tasks, hallucinating texts that are semantic but misaligned with the image is an important problem that can cause performance degradation and user frustration. Hence, the presented issue is important, and the associated benchmark contributed by the author(s) will be very useful for future research.

**Proposed method is effective in mitigating hallucination and improving scene-text-based tasks**: The proposed method is effective in targeting the presented semantic hallucination problem, and also improves some benchmarks focusing on scene-text tasks.

**Novel technical ideas in proposed method are potentially generalizable**: There are some novel ideas in the Glimpse-Refocus strategy that could be highly useful in not only mitigating semantic hallucination, but grounding useful regions/tokens in the problem in general.


**Weaknesses**:

**Unclear Technical Soundness of the Exploration**: It is unclear to me whether computing probability from intermediate states and the output weights is technically sound, given the output projection layer was trained only to handle the output of the final LLM layer. It appears that doing so will only isolate layers where clear concepts of output tokens have not yet been formed, instead of actually investigating the “hallucination source”.

**Impact on other capabilities**: Similar to the above point, as the output projection layer is only trained to handle the output from the final layer, it is unclear to me how the proposed method of “Grounded Layer Correction” would affect the models’ capabilities, especially on knowledge-based capabilities that rely more on text-based reasoning. In my opinion, the proposed method provides shortcuts for more localized, lower-level detail knowledge to be considered directly by the final layers, which would potentially degrade higher-level, knowledge-based capabilities. It would be great if the author(s) can include more general benchmarks outside of scene-text tasks, which would be important in cases where the LLM needs to handle not only scene-text-based tasks but also general tasks (hence cannot determine ahead of time whether the proposed method should be applied).

**Limited evaluation of the effectiveness of individual components of the framework** While the proposed method is evaluated end-to-end and has shown some improvements in downstream benchmarks, it would be helpful to understand the performance of individual components in the framework. In particular, the effectiveness of ZoomText can be measured with datasets where the ground-truth areas of interest are known. Similarly, the author(s) can also report the frequency where Grounded Layer Correction changes the output prediction. Reporting these metrics can significantly improve our understanding of the proposed method.

---

> ### Author Rebuttal · Authors · 2025-07-31
>
> # To reviewer LbB3
>
> We thank the reviewer for the detailed and encouraging feedback. We're pleased that you found our identification of the semantic hallucination issue in multi-modal large language models (MLLMs) to be both timely and significant, especially in light of the increasing use of MLLMs for text-rich tasks. We appreciate your recognition of the benchmark we contributed as a valuable resource for future research. We're also encouraged by your positive assessment of our method’s effectiveness in addressing semantic hallucination and enhancing performance on scene-text-focused benchmarks. Furthermore, we are grateful for your appreciation of the novel ideas behind our Glimpse-Refocus strategy and your insight that these ideas may generalize to broader applications involving grounded reasoning and alignment in vision-language tasks.
>
> Below, we respond to each Weakness (W) or Question (Q) with a corresponding Answer (A):
>
> ## [W1/Q1]: Unclear Technical Soundness of the Exploration
>
> [A1]: Thank you for your insightful question. First, we would like to clarify that the hidden states from intermediate layers are **not merely underdeveloped representations.** In fact, prior work has shown that intermediate layers in large language models can offer better representations for downstream tasks. For example, [1] concludes that intermediate layers provide stronger embeddings for natural language understanding tasks (see Sec. 4.1). This insight motivates our exploration in MLLMs.
>
> Building on this hypothesis, our experiments (Table 1) demonstrate that incorporating intermediate representations significantly improves grounding and reduces hallucination.
>
> That said, we acknowledge the potential mismatch between the intermediate hidden states and the output projection layer, since the model was only trained with final-layer outputs. To address this, we conduct an ablation study (Table 9), which shows that **assigning large weights** to intermediate layers can hurt performance, likely due to semantic disruption. However, **a moderate fusion weight preserves the semantic capacity of the final layer while benefiting from improved visual grounding** in intermediate layers.
>
>
> ## [W2/Q2]: Impact on other capabilities.
>
> [A2]: Thank you for your thoughtful comment. We would like to clarify that our proposed method, Grounded Layer Correction (GLC), is specifically designed to mitigate hallucination in scene text spotting and understanding by injecting localized visual grounding signals. However, this design does not compromise the model’s **general reasoning or knowledge-based capabilities.**
>
> As shown in Table 5, we evaluated our method on several general-purpose benchmarks beyond scene text. Still, we observed **minor improvements or comparable performance,** demonstrating that our method preserves the model’s broader abilities.
>
> To further validate this, we conducted **additional ablation studies on A-OKVQA [2],** a challenging benchmark that requires commonsense and world knowledge. Results show that applying our method **does not degrade the model’s performance,** confirming that GLC does not interfere with higher-level semantic reasoning.
>
> We do agree that our method does not significantly boost performance on knowledge-intensive tasks, primarily because it is **training-free** and does not introduce new external knowledge. In future work, we plan to explore combining hallucination mitigation with external knowledge grounding to further enhance general reasoning while preserving robustness in scene text understanding.
>
>
> | Model                 | A-OKVQA |
> |-----------------------|---------|
> | Qwen2.5-VL            | 85.2    |
> | Qwen2.5-VL + Ours     | 85.6    |
> | Mini-Monkey           | 77.7    |
> | Mini-Monkey + Ours    | 77.9    |
>
>
> ## [W3/Q3]: Limited evaluation of the effectiveness of individual components of the framework
>
> [A3]: Thank you for the valuable suggestion. We agree that evaluating the effectiveness of each individual component would provide deeper insight into our framework.
>
> **ZoomText Evaluation:**
>
> To assess the localization accuracy of ZoomText, we manually annotated 100 samples across TextHalu-Bench, TextVQA, and ST-VQA. Each sample was labeled with quadrilateral bounding boxes indicating the ground-truth text regions of interest. Since, to our knowledge, there is no existing benchmark that simultaneously offers both VQA annotations and dense spatial annotations, this manual annotation was necessary.
>
> We compare the following configurations on Qwen2.5-VL to measure the impact of each component in ZoomText:
>
> (1) Baseline: Selects top-k tokens based on visual attention from the final LLM layer.
>
> (2) + Glimpse: Incorporates query-to-image attention (Eq. 4) for token selection.
>
> (3) + Refocus: Further filters irrelevant tokens based on relative attention dynamics (Eq. 5).
>
>
> We compute the average IoU between the predicted tokens and ground-truth bounding boxes. The results show that ZoomText significantly improves text region localization, demonstrating its effectiveness in focusing on relevant visual regions.
>
>
> **Grounded Layer Correction (GLC)**:
>
> As for GLC, it is applied uniformly across all samples during decoding—there is **no conditional switching**. Therefore, we cannot meaningfully report a frequency of invocation. However, its effectiveness has been validated through extensive experiments showing consistent hallucination mitigation and semantic preservation (e.g., Table 1 and Table 5), confirming its utility as a general decoding enhancement.
>
> We will include these additional analyses and experimental results in the revised version of the paper.
>
>
>
> | Method      | Mini-Monkey | Qwen2.5-VL |
> |-------------|-------------|------------|
> | Baseline    | 42.3        | 46.1       |
> | + Glimpse   | 45.9        | 48.9       |
> | + Refocus   | 47.8        | 52.7       |
>
>
>
> [1] Does Representation Matter? Exploring Intermediate Layers in Large Language Models [NeurIPS 2024 workshop]
>
> [2] A-OKVQA: A Benchmark for Visual Question Answering using World Knowledge [ECCV 2022]

---

> ### Author Response · Authors · 2025-08-06
> **Seek Further Response**
>
> Dear Reviewer LbB3,
>
> Thank you for your support and helpful comments. We've tried our best to address your concerns, and we hope our responses make sense to you. Importantly, we much value your comments and would be happy to discuss more. If you have any additional questions or open discussions, please don't be hesitant to leave more comments. We are always available at all time, to actively address any concerns or be prepared for more discussions.
>
> Your opinions are rather important for us to improve the work!
>
> Thank you!
>
> Sincerely,
>
> Authors

---

> ### Comment · Reviewer_LbB3 · 2025-08-06
>
> Thank you for the author(s) detailed responsees. All of my concerns were addressed and I am willing to raise my score based on the rebuttal.

---

> > ### Author Response · Authors · 2025-08-07
> > **Official Comment by Authors**
> >
> > We sincerely appreciate your time and effort throughout the review. Thank you again for your valuable comments to improve the quality of our manuscript!

---

### Note · Authors · 2025-08-12

Dear Reviewers and AC,

We would like to express our sincere gratitude for your careful reviews and constructive feedback, which have substantially contributed to improving the quality of our work!

We are pleased that all the reviewers recognized the key strengths of our paper, including:

1. The presentation of an important issue (**semantic hallucination**), which is a timely and significant research problem.
2. The proposed method is well-motivated and effective in mitigating the hallucination problem.
3. The proposed benchmark is a huge contribution to the community.
4. Relevant technical ideas can be useful in general-purpose domains.

We greatly appreciate the opportunity to address the reviewers’ concerns during the rebuttal stage. In particular:

1. **Reviewer lbB3** — We clarified the technical soundness of the proposed method and complemented experiments on other knowledge-based benchmarks, as well as explored the individual components of the framework.
2. **Reviewer 1eg9** — We explained the expression ambiguity and complemented more experiments on OCRBench.
3. **Reviewer 6wHJ** — We discussed the main motivation and complemented experiments on the hyperparameter settings and efficiency.
4. **Reviewer wv1z** — We provided more quantitative results on MME and MME-VideoOCR, as well as ablation studies on the proposed method.

We are encouraged by the positive feedback from all reviewers, who indicated that most of their concerns have been addressed. Reviewer **lbB3** and Reviewer **1eg9** have promised to raise their scores, and the others maintain their original positive scores.

The corresponding results and analyses will be incorporated in the future version to further strengthen our paper. Once again, we sincerely thank you all for your valuable time and effort in reviewing our paper!

---

### Decision · Program_Chairs · 2025-09-17

**Decision:**

Accept (poster)

**Comment:**

Final rating: 4: Borderline Accept/  4: Borderline Accept/ 4: Borderline Accept/ 4: Borderline Accept. The paper tackles semantic hallucination in LMMs on ambiguous scene text, finding that layers with stronger attention to text hallucinate less. It proposes a training-free framework—ZoomText for coarse-to-fine text region focus and Grounded Layer Correction to decode with hallucination-resistant layers—and introduces TextHalu-Bench (1,730 samples) for evaluation. Experiments show effective hallucination mitigation and strong scene-text spotting/understanding performance.

Reviewers praise the timely problem, sound motivation, and the benchmark’s community value, but questioned (pre-rebuttal) the technical soundness of using intermediate-layer logits with the final head (and y_hal/y_gt in an autoregressive setup), possible side effects on broader capabilities, limited component attribution, overstated causality, reliance on baseline OCR (a corrective rather than capability-adding method), heuristic assumptions, and missing OCR benchmarks (OCRBench, MME/MME-VideoOCR).

After the rebuttal, reviewers report their concerns were satisfactorily addressed. The ACs concur that the problem about semantic hallucination is important, the method of the Glimpse-Refocus strategy is effective and the TextHalu-Bench benchmark is a substantial contribution. Therefore Acs recommend acceptance